# Polynomial Neural Fields
# for Subband Decomposition and Manipulation

**Guandao Yang**[*]
Cornell University

**Sagie Benaim**[*]
University of Copenhagen

**Varun Jampani**
Google Research

**Kyle Genova**
Google Research

**Jonathan T. Barron**
Google Research

**Thomas Funkhouser**
Google Research

**Bharath Hariharan**
Cornell University

**Serge Belongie**
University of Copenhagen

## Abstract

Neural fields have emerged as a new paradigm for representing signals, thanks to their ability to do it compactly while being easy to optimize. In most applications, however, neural fields are treated like black boxes, which precludes many signal manipulation tasks. In this paper, we propose a new class of neural fields called polynomial neural fields (PNFs). The key advantage of a PNF is that it can represent a signal as a composition of a number of manipulable and interpretable components without losing the merits of neural fields representation. We develop a general theoretical framework to analyze and design PNFs. We use this framework to design Fourier PNFs, which match state-of-the-art performance in signal representation tasks that use neural fields. In addition, we empirically demonstrate that Fourier PNFs enable signal manipulation applications such as texture transfer and scale-space interpolation. Code is available at https://github.com/stevenygd/PNF.

## 1 Introduction

Neural fields are neural networks that take as input spatial coordinates and output a function of the coordinates, such as image colors [8, 56], 3D signed distance functions [3, 46], or radiance fields [39]. Recent works have shown that such representations are compact [15, 35, 51], allow sampling at arbitrary locations [56, 59], and are easy to optimize within the deep learning framework. These advantages enabled their success in many spatial visual computing applications including novel view synthesis [39, 41, 43, 55, 68] and 3D reconstruction [17, 27, 47, 55, 56]. Most recent neural field based methods, however, treat the network as a black box. As such, one can only obtain information by querying it with spatial coordinates. This precludes the applications where we want to change the signal represented in a neural field. For example, it is difficult to remove high frequency noise or change the stationary texture of an image.

One way to enable signal manipulation is to decompose a signal into a number of interpretable components and use these components for manipulation. A general approach that works across a wide range of signals is to decompose them into frequency subbands [1, 7, 54]. Such decompositions are studied in the traditional signal processing literature, e.g., the use of Fourier or Wavelet transforms of the spatial signal. These transformations, however, usually assume the signal is densely sampled

---

[*]Equal contribution. Part of this work was done while Guandao was a student researcher at Google.

36th Conference on Neural Information Processing Systems (NeurIPS 2022).

in a regular grid. As a result, it is non-trivial to generalize these approaches to irregular data (e.g., point clouds). Another shortcoming of these transforms is that they require a lot of terms to represent a signal faithfully, which makes it difficult to scale to signals of more than two dimensions, such as in the case of light fields. Interestingly, these are the very problems that can be solved by neural fields, which can represent irregular signals compactly and scale easily to higher dimensions. This leads to the central question of this paper: can we incorporate the interpretability and controllability of classical signal processing pipelines to neural fields?

Our goal is to design a new class of neural fields that allow for precise subband decomposition and manipulation as required by the downstream tasks. To achieve that, our network needs to have the ability to control different parts of the network to output signal that is limited by desirable subbands. At the same time, we want the network to inherit the usual advantages of neural fields, namely, being compact, expressive, and easy to optimize. The most relevant prior works with related aims are Multiplicative Filter Network (MFN) [19] and BACON [32]. While MFNs enjoy the advantages of neural fields and are easy to analyze, they do not use this property to control the network's output for subband decomposition. BACON [32] extends the MFN architecture to enforce that its outputs are upper-band limited, but it lacks the ability to provide subband control beyond upper band limits. This hinders BACON's applicability to tasks that requires more precise control of subbands, such as manipulating stationary textures (shown in Sec. 4.2).

To address these issues, we propose a novel class of neural fields called **polynomial neural fields (PNFs)**. PNF is a polynomial neural network [10] evaluated using a set of basis functions. PNFs are compact, easy to optimize, and can be sampled at arbitrary locations as with general neural fields. Moreover, PNFs enjoy interpretability and controllability of signal processing methods. We provide a theoretical framework to analyze the output of PNFs. We use this framework to design the Fourier PNF, whose outputs can be localized in the frequency domain with both upper and lower band limits, along with orientation specificity. To the best of our knowledge, this is the first neural field architecture that can achieve such a fine-grained decomposition of a signal. Empirically, we demonstrate that Fourier PNFs can achieve subband decomposition while reaching state-of-the-art performance for signal representation tasks. Finally, we demonstrate the utility of Fourier PNFs in signal manipulation tasks such as texture transfer and scale-space interpolation.

## 2 Related Works

Our method are built on three bodies of prior works: signal processing, neural fields, and polynomial neural networks. In this section, we will focus on the most relevant part of these prior works. For further readings, please refer to Orfanidis [45] for signal processing, Xie et al. [66] for neural fields, and Chrysos et al. [12] for polynomial neural networks.

**Fourier and Wavelet Transforms.** In traditional signal processing pipeline, one usually first transforms the signal into weighted sums of functionals from certain basis before manipulating and analyzing the signal [45]. The Fourier and Wavelet transformation are most relevant to our work. In particular, prior works has leveraged Fourier and Wavelet transforms to organize image signal into meaningful and manipulable components such as the Laplacian [7] and Steerable [54] pyramids. In our paper, we analyze the signal in terms of the basis functions studied by Fourier and Wavelet transforms. Our manipulable components are also inspired by the subband used in Steerable Pyramid [53]. While this signal processing pipeline is very interpretable, it's non-trivial to make it work on irregular data because these transformations usually assume the signal to be densely sampled in a regular grid. In this paper, we tries to combines the interpretability of traditional signal processing pipeline with the merits of neural fields, which is easy to optimize even with irregular data.

**Neural Fields.** Neural Fields are neural networks that maps spatial coordinate to a signal. Recent research has shown that neural fields are effective in representing a wide variety of signals such as images [56, 58], 3D shapes [3, 14, 37, 46, 67], 3D scenes [17, 24, 27, 47, 55]. and radiance fields [4, 5, 6, 20, 28, 31, 33, 34, 36, 39, 40, 41, 43, 44, 48, 50, 55, 57, 63, 68, 69, 70, 71]. However, neural fields typically operate as black boxes, which hinders the application of neural fields to some signal decomposition and manipulation tasks as discussed in Sec. 1. Recent works have to alleviate such issue by designing network architecture that are partially interpretable. A common technique is to encode the input coordinate with a positional encoding where one can control spectrum properties such as the frequency bandwidth inputting into the network [4, 39, 56, 59, 72]. But these positional

encodings are passed through a black-box neural networks, making it difficult to analyze properties of the final output. The most relevant works is BACON [32], which propose an initialization schema for multiplicative filter networks (MFNs) [18] that ensures the output to be upper-limited by certain bandwidth. This work generalizes BACON in two ways. First, our theory can be applied to a more general set of basis function and network topologies. Second, our network enables more precise subband controls, which include band-limiting from above, below, and among certain orientation.

**Polynomial Neural Networks.** Polynomial neural networks (PNNs) are generally referred to neural networks composed of polynomials [12]. The study of PNNs can be dated back to higher-order boltzmann machine [52] and Mapping Units [25]. Recently, research has shown that PNNs can train very strong generative models [10, 11, 13, 23, 65] and recognition models [26, 64]. The empirical success has also followed with deeper theoretical analysis. For example, [13] reveal how PNNs' architecture relates to polynomial factorization. Kileel et al. [29] and Choraria et al. [9] studies the expressive power of PNNs. Our work establish the connection between PNNs and many neural fields such as MFN [19] and BACON [32]. We further extends polynomial neural networks by evaluating the polynomial with a set of basis functions such as the Fourier basis.

## 3 Method

In this section, we will provide a definition for polynomial neural fields (Sec. 3.1). From this definition, we derive a theoretical framework to analyze their outputs in terms of subbands (Sec. 3.2). We then use this framework to design Fourier PNFs, a novel neural fields architecture to represent signals as a composition of fine-grain subbands in frequency spaces (Sec. 3.3).

### 3.1 Polynomial Neural Fields

Recall that we would like to maintain the merits of the neural field representation while adding the ability to partition it into analyzable components. As with all neural networks, to guarantee expressivity, we want to base our neural fields on function compositions [49]. At the same time, we want our neural fields to be interpretable in terms of a set of basis functions that have known properties, as in the signal processing literature. We propose the following class of neural fields:

**Definition 3.1** (PNF). *Let $\mathcal{B}$ be a basis for the vector space of functions for $\mathbb{R}^n \to \mathbb{R}$. A Polynomial neural field of basis $\mathcal{B}$ is a neural network $f = g_L \circ \cdots \circ g_1 \circ \gamma$, where $\forall i, g_i$ are finite degree multivariate polynomials, and $\gamma : \mathbb{R}^n \to \mathbb{R}^d$ is a $d$-dimensional feature encoding using basis $\mathcal{B}$: $\gamma(x) = [\gamma_1(x) \ \ldots \ \gamma_d(x)]^T, \gamma_i \in \mathcal{B}, \forall i$.*

This definition allows a rich design space that subsumes several prior works. For example, MFN [19] and BACON [32] can be instantiated by setting $g_i$ to be either the linear layer or a masked multiplication layer. Similarly, if we set the basis to be $\mathcal{B} = \{x^n\}_{n \in \mathbb{N}}$, then we can show architectures proposed in $\Pi$-Net [13] are a subclass of PNF. This rich design space can potentially allow us to tailor the architecture to the application of interests, as demonstrated later in Sec. 3.3 and Sec. 4.3. Moreover, such rich space also contains many expressive neural networks, as shown by both the prior works [10, 13, 19, 32] and by our experiment in Sec. 4.1.

Furthermore, as long as the span of the basis is closed under multiplication, PNF yields a linear combination of basis functions and is thus easy to analyze:

**Theorem 1.** *Let $F$ be a PNF with basis $\mathcal{B}$ s.t $\forall b_1, b_2 \in \mathcal{B}, b_1(x)b_2(x) = \sum_{i \in I} a_i b_i(x), |I| < \infty$. Then the output of $F$ is a finite linear sum of the basis functions from $\mathcal{B}$.*

Many commonly used bases, such as Fourier, Gabor, and spherical harmonics, all satisfy this property. Please refer to the supplementary for proofs for a variety of different bases.

### 3.2 Controllable Subband Decomposition

Theorem 1 is not enough to control or manipulate the signal represented by the network, because any single neuron in the network may potentially be working with an arbitrary set of basis functions. This is a problem if we want to manipulate the signal through these neurons. For example, if we want to discard the contributions of some high-frequency components in the Fourier basis, we cannot decide which neurons to discard if each of them are contributed to a variety of frequency bands.

To allow better manipulation, we use the notion of *subbands* from traditional frequency domain analysis [62]. In the most general sense, a subband is simply a *subset* of the basis. Manipulations such as smoothing or sharpening can then be done by discarding or enhancing the contributions of one or more such subbands. One way to instantiate these ideas with a PNF is to represent the signal as a sum of different PFNs, each of which is limited to only a specific subband; then one can manipulate these component PFNs separately.

**Definition 3.2.** *A PNF $F$ of basis $\mathcal{B}$ is limited by a subband $S \subset \mathcal{B}$ if $F$ is in the span of $S$.*

A key challenge is to construct a subband limited PNF for certain subbands. To this end, we need to understand how the subband-limited PNF transforms under different network operations. Fortunately, for PNFs we only need to study two types of operations: multiplication and addition. To this end, we need the notion of a *PNF-controllable set of subbands*:

**Definition 3.3** (PNF-controllable Set of Subbands)**.** *$\mathcal{S} = \{S_\theta | S_\theta \subset \mathcal{B}\}_\theta$ is a PNF-Controllable Set of Subbands for basis $\mathcal{B}$ if (1) $S_{\theta_1} \cup S_{\theta_2} \in \mathcal{S}$ and (2) there exists a binary function $\otimes : \mathcal{S} \times \mathcal{S} \to \mathcal{S}$ such that if $b_1 \in S_{\theta_1}, b_2 \in S_{\theta_2} \implies b_1 b_2 \in S_{\theta_1} \otimes S_{\theta_2}$.* [2]

Intuitively, a PNF-controllable set of subbands lends some *predictability* to what happens if two PNFs, limited to different subbands, are put together into a larger PNF:

**Theorem 2.** *Let $\mathcal{S}$ be a PNF-controllable set of subbands of basis $\mathcal{B}$ with binary relation $\otimes$. Suppose $F_1$ and $F_2$ are polynomial neural fields of basis $\mathcal{B}$ that maps $\mathbb{R}^n$ to $\mathbb{R}^m$. Furthermore, suppose $F_1$ and $F_2$ are subband limited by $R_1 \in \mathcal{S}$ and $R_2 \in \mathcal{S}$. Then we have the following:*

1. *$W_1 F_1(x) + W_2 F_2(x)$ is a PNF of $\mathcal{B}$ limited by subband $R_1 \cup R_2$ with $W_1, W_2 \in \mathbb{R}^{m \times n}$;*

2. *$F_1(x)^T W F_2(x)$ is a PNF of $\mathcal{B}$ limited by subband $R_1 \otimes R_2$ with $W \in \mathbb{R}^{m \times m}$.*

Many structures used in the signal processing literature, such as the Steerable Pyramid, use subbands that are PNF-controllable [7, 54]. Please refer to the supplementary for the derivations of PNF-controllable sets of subbands. In the following sections, we will focus on using Fourier bases and show how to build PNFs that instantiate subband manipulation efficiently.

### 3.3 Fourier PNF

In this section, we demonstrate how to build a PNF that can decompose a signal into frequency subbands in the following three steps. First, we identify a PNF-controllable sets of subband for the Fourier basis. Second, we choose a finite collection of subbands for the PNF to output, and organize them into controllable sets. The final step is to instantiate the PNF compactly using Theorem 2.

#### 3.3.1 Controllable Subband Decomposition for Fourier Space

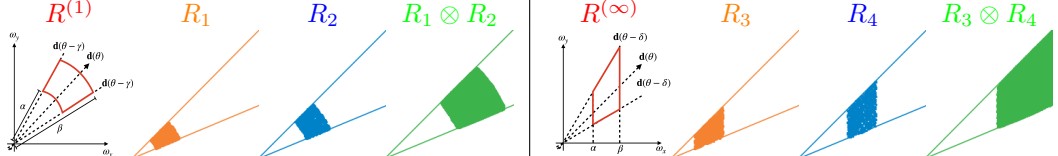

Figure 1: Illustration of PNF-controllable subbands in Fourier space. Left: 2-norm series of subbands; Right: infinite-norm series of subbands.

The Fourier basis can be written as: $b_{\boldsymbol{\omega}}(\mathbf{x}) = \exp(i\boldsymbol{\omega}^T \mathbf{x})$, where $\boldsymbol{\omega}, \mathbf{x} \in \mathbb{R}^d$ and $i$ is the imaginary unit. It's easy to see that the fourier basis is complete under multiplication, which satisfies the condition for Theorem 1: $\exp(i\boldsymbol{\omega}_1^T \mathbf{x}) \exp(i\boldsymbol{\omega}_2^T \mathbf{x}) = \exp(i(\boldsymbol{\omega}_1 + \boldsymbol{\omega}_2)^T \mathbf{x})$.

Now we need to divide frequency space of $\boldsymbol{\omega} \in \mathbb{R}^d$ into subbands that are easy to manipulate and meaningful for downstream tasks. We define the subband following Simoncelli and Freeman [54]. Formally, frequency space is decomposed into following sectors:

$$R^{(p)}(\alpha, \beta, \mathbf{d}, \gamma) = \left\{ \boldsymbol{\omega} | \alpha \leq \|\boldsymbol{\omega}\|_p \leq \beta, \|\mathbf{d}\| = 1, \boldsymbol{\omega}^T \mathbf{d} < \gamma \|\boldsymbol{\omega}\| \right\}, \tag{1}$$

---

[2]It's possible prove Theorem 2 with a more relax version of Definition 3.3: $b_1 b_2 \in Span(S_{\theta_1} \otimes S_{\theta_2})$.

where $p$ describe which norm do we choose to describe the frequency bands. Intuitively, $\alpha$ defines the lower band limits and $\beta$ defines the upper band limits. The vector $\mathbf{d}$ defines the orientation of the subband and $\gamma$ defines the angular width of the subband. Fig. 1 provides illustrations of $R^{(p)}$.

This definition of subband allows us to organize them into controllable sets. For example, we can show that the following sets of subbands are controllable:

**Theorem 3.** *Let $\mathcal{S}$ be a set of subbands defined as $\mathcal{S} = \{R^{(2)}(\alpha, \beta, \mathbf{d}, \gamma) | \forall 0 \le \alpha \le \beta\}$. If $|\gamma| < \frac{\pi}{4}$, then $\mathcal{S}$ is PNF-controllable. Specifically, if $\boldsymbol{\omega}_1 \in R_1^{(2)}(\alpha_1, \beta_1) \in \mathcal{S}$, and $\boldsymbol{\omega}_2 \in R_2^{(2)}(\alpha_2, \beta_2) \in \mathcal{S}$, then $b_{\boldsymbol{\omega}_i} b_{\boldsymbol{\omega}_i} = b_{\boldsymbol{\omega}_3}$ implies that $\boldsymbol{\omega}_3 \in R^{(2)}(\sqrt{\cos(2\gamma)}(\alpha_1 + \alpha_2), \beta_1 + \beta_2) \in \mathcal{S}$.*

This theorem captures the intuition that the multiplication of two waves of similar orientations creates high-frequency waves at that orientation. It allows us to predict the spectrum properties of the output of the network when knowing the spectrum properties of the inputs. Fig. 1 provides illustrations of how two 2D subbands interact under multiplications.

### 3.3.2 Subband Tiling

In order to represent different signals, our networks need to be able to leverage basis functions with different orientations and within different bandwidths. With that said, we need to choose a set of PNF-controllable subbands to cover all basis functions we want to use. For example, we can tile the space with the set of controllable subbands in Theorem 3 in the following way:

$$\mathcal{T}_{circ} = \{S_{ij} = R^{(2)}(b_i, b_{i+1}, \mathbf{d}(\theta_j), \delta) | b_1 \le \cdots \le b_n, \theta_j = j\delta, \delta = \frac{\pi}{m}, 1 \le j \le 2m\}, \quad (2)$$

where $\mathbf{d}(\theta) = [\sin(\theta), \cos(\theta)]^T$ denotes unit vector rotate with angle $\theta$, $\delta = \frac{\pi}{m}$, and $m$ sufficiently small to allow the application of Theorem 3.

For 2D images, the region of interest is $[-N, N]^2$ where $N$ is the bandwidth determined by the Nyquist Sampling Theorem [42]. To tile this rectangular region well without introducing unnecessary high-frequency details, we will use pesudo polar coordinate grids, which divide the space into vertical or horizontal sub-regions and tile those sub-regions according to $l$-$\infty$ norm. Formally, the 2D pseudo polar coordinate tiling can be written as:

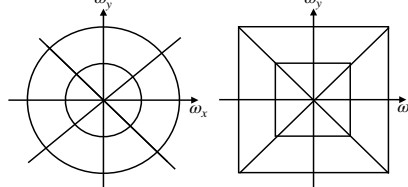

Figure 2: Left: $\mathcal{T}_{circ}$; Right: $\mathcal{T}_{rect}$.

$$\mathcal{T}_{rect} = \{S_{ij} = R^{(\infty)}(b_i, b_{i+1}, \mathbf{d}(\theta_j), \delta) | b_1 \le \cdots \le b_n, \theta_j = j\delta, 1 \le j \le 2m, j \ne m\}. \quad (3)$$

Note that we exclude certain regions to avoid having a subband to include orientation at $\frac{\pi}{4}$ and $\frac{3\pi}{4}$ in order for Theorem 3 to generalize to such tiling. Fig. 2 contains an illustration of these two types of tiling. We show detailed derivation in the supplementary.

Tiling the spectrum space with subbands from $\mathcal{T}_{circ}$ or $\mathcal{T}_{rect}$ allows us to organize information in a variety of meaningful ways. For example, this set of subbands can be grouped into different cones $\{S_{ij}\}_{j=1}^{2m}$. Each cone corresponds to a particular orientation of the signal. Alternatively, we can also organize this set of subbands into different rings $\{S_{ij}\}_{i=1}^{n}$, which corresponds to the decomposition of an image into the Laplacian Pyramid.

### 3.3.3 Network architecture

We now have decided on the set of subbands we want to produce by PNF. We want to design the final Fourier PNF as an ensemble of subband limited PNFs: $F(\mathbf{x}) = \sum_j \sum_i O_{ij} F_{ij}(\mathbf{x})$, where $F_{ij}(\mathbf{x}) : \mathbb{R}^n \to \mathbb{R}^h$ is a PNF that's limited with subband $S_{ij}$ defined in Eq. (3), and $O_{ij} \in \mathbb{R}^{h \times m}$ aggregates the output signals together. One way to achieve this is to naively define $F_{ij}$ as a two layers PNF the feature encoding layer to include only the basis functions in the subband $S_{ij}$ followed by a linear layer. However, such an approach fails to achieve good performance without a huge number of trainable parameters. Alternatively, we leverage Theorem 2 to factorize $F$:

$$F(\mathbf{x}) = \sum_j F_j(\mathbf{x}), \quad F_j(\mathbf{x}) = \sum_{k=1}^{n} G_j(\mathbf{x}, b_i, b_i) W_{jk} Z_{j,k}(\mathbf{x}), \quad (4)$$

$$Z_{j,1}(\mathbf{x}) = G_j(\mathbf{x}, 0, \Delta_1), \quad Z_{j,k}(\mathbf{x}) = G_j(\mathbf{x}, 0, \Delta_k) W_i Z_{j,k-1}(\mathbf{x}), \quad (5)$$

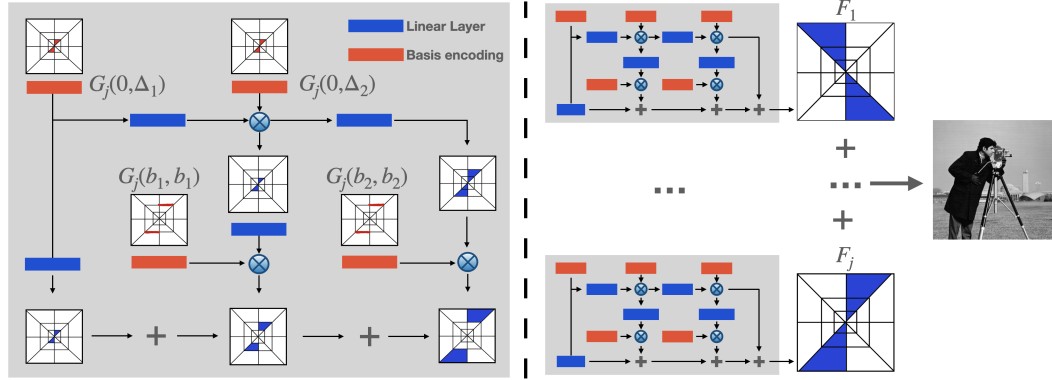

Figure 3: Illustration of Fourier PNF architecture. Fourier PNF is an ensemble model. The final result is summed over a series of PNFs $F_j$, whose structure is shown on the left side of the figure.

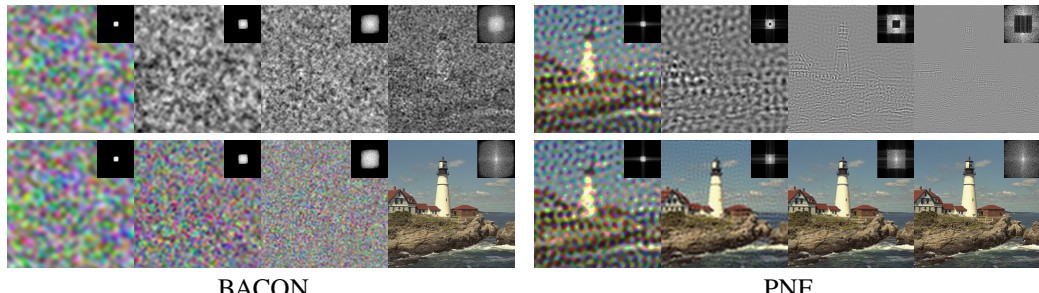

BACON                                                                PNF

Figure 4: BACON: The bottom row shows the output of each layer which is upper band limited. The top row (columns 2-4) shows the difference between the output of a given layer and the one before it. PNF: The top row shows the output of each layer which is both upper and lower band limited. The bottom row (columns 2-4) shows the addition of the output at a given layer and the one before it.

where $G_j(\mathbf{x}, a, b)$ is subband limited in $R^{(\infty)}(a, b, d(\theta_j), \delta)$ and $\Delta_k = b_k - b_{k-1}$. This network architecture of $F_i$ is illustrated in Fig. 3. We instantiate this architecture by setting $G_j(\mathbf{x}, a, b)$ into a linear transform of basis sampled from the subband to be limited:

$$G_j(\mathbf{x}, a, b) = W_i \gamma_j(\mathbf{x}), \gamma_j \in R^{(\infty)}(a, b, d(\theta_j), \delta)^d, W_i \in \mathbb{R}^{h \times d}, \tag{6}$$

where $h$ and $d$ is the dimension for the output and the feature encoding. We provide additional implementation details in the supplementary.

## 4   Results

We demonstrate the applicability of our framework along three different axes: (*Expressivity*) For a number of different signal types, we demonstrate our ability to fit a given signal. We observe that our method enjoys faster convergence in terms of the number of iterations. (*Interpretability*) We visually demonstrate our learned PNF, whose outputs are localized in the frequency domain with both upper band-limit and lower-band limit. (*Decomposition*) Finally, we demonstrate the ability to control a PNF on the tasks of texture transfer and scale-space representation, based on our subband decomposition. For all presented experiments, full training details and additional qualitative and quantitative results are provided in the supplementary.

Table 1: Image Fitting on the DIV2K dataset.

|  | PSNR | SSIM | # Params |
|---|---|---|---|
| RFF | 28.72 | 0.834 | 0.26M |
| SIREN | 29.22 | 0.866 | 0.26M |
| BACON | 28.67 | 0.838 | 0.27M |
| BACON-L | 29.44 | 0.871 | 0.27M |
| BACON-M | 29.44 | 0.871 | 0.27M |
| PNF | **29.47** | **0.874** | 0.28M |

Table 2: 3D shape fitting. CD is Chamfer Distance ($\times 10^6$)

|  | CD | F-score | # Params |
|---|---|---|---|
| SIREN | 9.00 | 99.76% | 0.53M |
| BACON | 2.60 | 99.84% | 0.54M |
| BACON-L | 2.60 | 99.85% | 0.54M |
| BACON-M | 2.61 | 99.85% | 0.54M |
| PNF | **2.25** | **99.97%** | 0.59M |

Table 3: NeRF Fitting for $64^2$ resolution. A comparison between PNF and bacon is shown for PSNR and SSIM at 300 and 500 epochs. as well as corresponding number of parameters used.

|  |  | 1x | 1/2x | 1/4x | 1/8x | Avg |
|---|---|---|---|---|---|---|
| 300 epochs | BACON | 28.07/0.936 | 29.95/0.943 | 30.75/**0.939** | 31.37/**0.927** | 30.04/**0.936** |
| PSNR/SSIM | PNF | **29.89/0.937** | **31.16/0.946** | **32.11**/0.934 | **32.00**/0.920 | **31.29**/0.934 |
| 500 epochs | BACON | 28.51/0.932 | 30.77/0.941 | **31.74/0.940** | **32.10/0.928** | 30.78/0.935 |
| PSNR/SSIM | PNF | **29.33/0.950** | **31.19/0.950** | 31.48/0.936 | 31.37/0.926 | **30.84/0.940** |
| # Params | BACON | 0.54M | 0.41M | 0.27M | 0.14M | 0.34M |
|  | PNF | 0.46M | 0.34M | 0.23M | 0.12M | 0.29M |

## 4.1 Expressivity

We demonstrate that a PNF is capable of representing signals of different modalities including images, 3D signed distance fields, and radiance fields. We compare our Fourier PNF with state-of-the-art neural field representations such as BACON [32], SIREN [56], and Random Fourier Features [59].

**Images** Following BACON [32], we train a PNF and the baselines to fit images from the DIV2K [2] dataset. During training, images are downsampled to $256^2$. All networks are trained for 5000 iterations. At test time, we sample the fields at $512^2$ and compare with the original resolution images.

Different from other networks, BACON is supervised with the training image in all output layers. For fair comparison, we also include two BACON variants, which are only supervised either at the last output layer ("BAC-L") or using the average of all output layers ("BAC-M"). We report the PSNR and Structural Similarity (SSIM) scores of these methods in Tab. 1. Fourier PNF improves on the performance of the previous state-of-the-arts.

In addition to its expressivity, PNF is also able to localize signals in different regions. Specifically, Fourier PNF decomposes an image into subbands in the frequency domain. For example, Fig. 4 shows that the output branches of the Fourier PNF correspond to specific frequency bands. Note that such decomposition forms for all layers while training is only performed for the last layer of the PNF. Such subband control is reminiscent of the traditional signal processing analogue of Laplacian Pyramids [7]. Note that each layer of the PNF is guaranteed to be both lower and upper frequency band limted, while this is not achievable by BACON. For comparison Fig. 4 shows the corresponding result of training BACON only for the last layer. In Sec. 4.2 and Sec. 4.3 we demonstrate how to leverage such fine-grain localization for texture transfer and scale space interpolation.

**3D Signed Distance Field** One advantage of neural fields is their ability to represent irregular data such as 3D point cloud or signed distance fields with high fidelity. We demonstrate that PNFs can represent 3D shapes via signed distance fields expressively. We follow BACON's experimental setting to fit a range of 3D shapes from the Stanford 3D scanning repository [60] (a slightly different normalization is used, see supplementary for details). During training, we sample $10k$ oriented points from the ground truth surface and perturb them with noise to compute an estimate of SDF as ground truth. All models are trained with reconstruction loss for the same number of iterations. A quantitative comparison of the different methods is reported in Tab. 2. We observe that PNF achieves

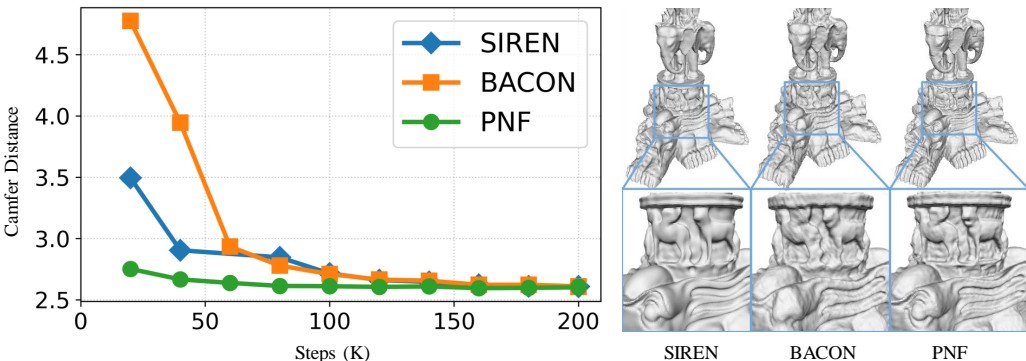

Figure 5: LHS: Convergence time in terms of number of steps/iterations for the *Thai Statue* model [60]. The x-axis shows the number of steps (K) and the y-axis shows the validation Chamfer Distance ($\times 10^6$). RHS: Qualitative comparison for the *Thai Statue* model [60].

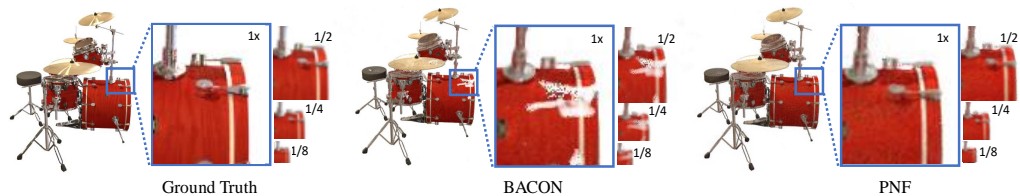

Figure 6: Qualitative comparison of neural radiance fields for the *Drums* scene given in [4] at the 300 epochs. The results suggests that our model can achieve better quality in early iterations.

slightly higher fitting quality than other methods. Next, we investigate convergence behavior in Fig. 5 (Left) on the *Thai Statue* model [60]. Here, we observe that PNFs converge much more quickly than SIREN and BACON while achieving comparable quality. In Fig. 5 (Right) we provide a qualitative comparison of our result, BACON, and SIREN.

**Neural Radiance Field**    In the context of neural radiance fields, we show that PNFs can represent signals in higher dimensions compactly and faithfully. We follow BACON's setting and train a PNF to model the radiance field of a set of Blender scenes [39], see supplementary for details. Specifically, the PNF outputs a 4D vector of RGB and density values. These values are used by a volumetric renderer proposed of NeRF [39] to produce an image, which is supervised with reconstruction loss. At test time, we use the same volumetric renderer to produce images from different camera poses and evaluate them with the known ground truth images. The results, in comparison to BACON, are presented in Tab. 3. A Qualitative comparison is also given in Fig. 6. Similarly to SDF, we are able to match or improve BACON's performance using about 60% of the total number of iterations. This can be potentially attributed to PNF's ability to disentangle coarse signal from finer one; and so at higher layers, BACON needs to relearn low-frequency details in the image while the PNF does not.

**Parameter Efficiency.**    One major advantage of using neural fields is that it can represent signal with high expressivity while remaining compact. In this section, we will demonstrate that PNF also retains this advantages. While choosing the hyperparameters for PNFs (e.g. hidden layer size) for the expressivity experiemnts, we make sure the PNF has a comparable number of parameters with the prior works. A comparison of the number of trainable parameters used for different model is included in Tab. 1-Tab. 3. We can see that PNFs are achieving comparable performance with the state-of-the-arts neural fields with roughly the same amount of trainable parameters.

**Training and Inference Time**    While we observe that Fourier PNF can converge in fewer iterations, but due to the ensemble nature of Fourier PNF, each forward and backward pass of PNF requires longer time to evaluate. This leads to the question, can we actually achieve faster convergence in wall time? We profile the training and inference time of the image fitting experiment in Tab. 4 for the

Table 4: A comparison of the training and inference time for image fitting of a cameramen image.

|  | Time(s)/Step | Time(s) to 36 PSNR | Final PSNR | Final SSIM |
|---|---|---|---|---|
| BACON | 0.16 | 177 | 37.45 | 97.33 |
| PNF | 0.64 | 96 | 37.45 | 97.44 |
| SIREN | 0.10 | 163 | 36.90 | 97.50 |
| RFF | 0.08 | 275 | 36.23 | 95.05 |

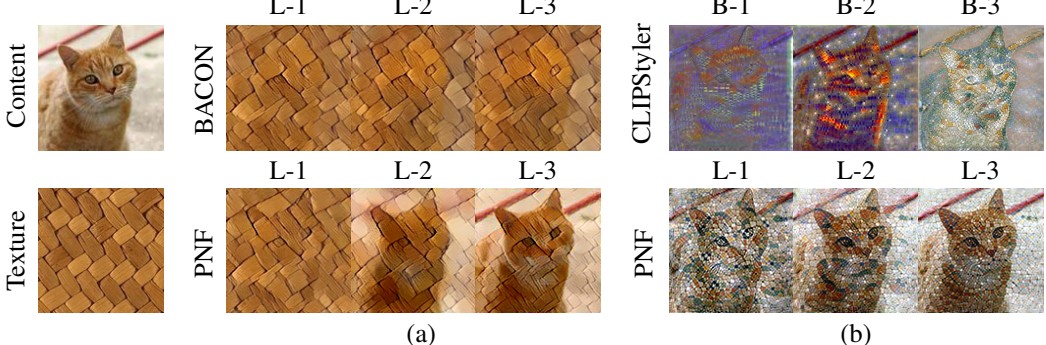

(a)           (b)

Figure 7: Texture transfer. (a). We optimize specific layers of the neural field. L-1 (layers 1-4), L-2 (layers 2-4), L-3 (layers 3-4). The texture contains stationary texture (in high frequency) and structured texture (in low frequency). As opposed to BACON, our method can isolate the stationary texture. (b) We consider the text based texture transfer objectives of CLIPStyler [30] for the cat content image. For PNF, we consider the text prompt "Mosaic" and apply the same layer-based optimization as in (a). For comparison, we apply CLIPStyler of "Low frequency mosaic" (B-1), "High frequency mosaic" (B-3) and "Mosaic" (B-3).

camera men image. The results show that it's possible for PNF to converge faster even in terms of clock time because PNFs can converge is drastically fewer number of steps.

## 4.2 Texture Transfer

Traditional approaches for image manipulation assume the input and output images are represented using a regular grid [16, 22]. Recent work has opted to use neural fields instead, for example allowing fine detailed texturing of 3D meshes [38]. Our formulation allows for an additional layer of control. In particular, due to our subband decompositionality, one can restrict the manipulation to particular subbands. The manipulation can then be driven by optimizing the PNF weights corresponding to those subbands, using various loss objectives.

We demonstrate this in the setting of texture transfer. We consider a Fourier PNF with four layers of the following frequency ranges (in Hz): (1) $[0, 8]$, (2) $[4, 16]$, (3) $[12, 32]$ and (4) $[28, 64]$. We consider a *content image* C of $128^2$ resolution. In the first stage, we train a network to fit C as in Sec. 4.1. In the second stage, we optimize only the parameters of specifics layers. To optimize these parameters, we query the network on a $128^2$ image grid producing image I. We then consider two sets of objectives: (a) Content and style loss objectives as given in [21]. (b), Text-based texture manipulation objectives as given by CLIPStyler [30]. See further details in the supplementary.

In Fig. 7, we illustrate the result of optimizing layers 1 to 4, 2 to 4 or 3 to 4, corresponding to frequency ranges [0, 64], [4, 64] and [12, 64] respectively. We consider a texture which contains both stationary texture in the high frequency range and structured texture in the low frequency range. As opposed to BACON, our method can isolate the stationary texture. As can be seen in Fig. 7(b), for text based manipulation, our method generates texture that is in the correct frequency range. For comparison, we consider CLIPStyler [30], with text prompts of "Low frequency mosaic" (B-1), "High frequency mosaic" (B-3) and "Mosaic" (B-3), resulting in non-realistic texturing. As can be seen, simply specifying the frequency in the text does not result in a satisfactory result.

### 4.3 Scale-space Representation

In many visual computing applications such as volumetric rendering, one is usually required to aggregate information from the neural fields using operations such as Gaussian convolution [4]. It is useful to model signals as a function of both the spatial coordinates and the scale: $f(\mathbf{x}, \Sigma) = \mathbb{E}_{\mathbf{x} \sim \mathcal{N}(\mathbf{x}, \Sigma)}[g(\mathbf{x})]$, where $g$ is the assumed ground truth signal. Existing works try to approximate the scale-space by using a black-box MLP with intergrated positional encoding, which computes the analytical Gaussian convolved Fourier basis functions [4, 61]. While such approaches demonstrate success in volumetric rendering applications which requires integrating different scales, they depend on supervision in multiple scales, possibly because their ability to interpolate correctly between scales is hindered by the black-box MLP.

In this section we want to demonstrate the PNF's ability to better model this scale space with limited supervision. Suppose the signal of interest can be represented by Fouier bases as $g(\mathbf{x}) = \sum_n \alpha_n \exp(\omega_i^T \mathbf{x})$, then we know analytically the Gaussian convolved version should be $f(\mathbf{x}, \Sigma) = \sum_n \alpha_n \exp(\omega_i^T \Sigma \omega_i) \exp(i\omega_i^T \mathbf{x})$. If we assume that our Fourier PNF can learn the ground truth representation well, then one potential way to achieve this is setting $\gamma(\mathbf{x}, \Sigma)_n = \exp(-\frac{1}{2}\omega_n^T \Sigma \omega_n) \exp(-i\omega_n^T \mathbf{x})$. We also multiply the output of $F_i$ (Eq. (4)) with a correction term addressing the error arises from missing the interference terms in the form of $\exp(-\frac{1}{2}\omega_i^T \Sigma \omega_j)$. We show in the supplementary how to derive and approximate these missing terms using Fourier PNF.

We test the effectiveness of Fourier PNF to learn the scale-space of a 2D image. In this application, the network is trained with the image signal in full resolution (finest scale). At test time, the network is asked to produce image with different scales and compared to the ground truth Gaus-

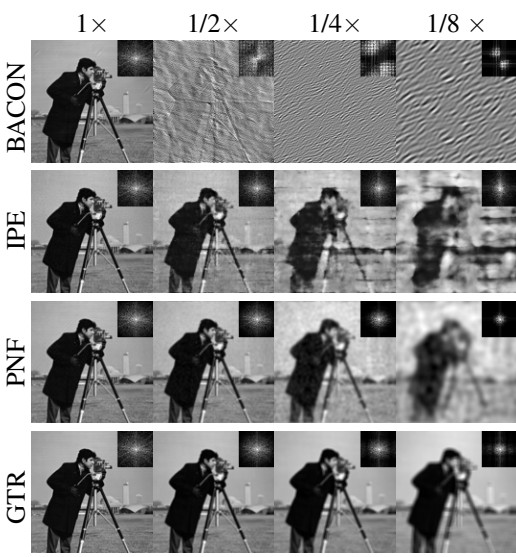

Figure 8: Scale-space representation. Networks are trained on full resolution image (1x) and test on the other reolutions. GTR is produced by applying Gaussian smoothing on (1x) image.

sian smoothed image. We compared our method with IPE [4] as well as BACON with IPE as filter function. The results are shown in Fig. 8. Our model can represent a signal reasonably well when testing with a lower resolution while other methods degrade more quickly.

**Limitations** Currently, the activation memory of PNF scales linearly in the number of subbands, and so interpretability and decompositiality gained by PNF comes at a "cost" of a larger memory footprint. Further, it is also non trivial to tile higher dimensional space with controllabel subbands.

## 5 Conclusion

We proposed a novel class of neural fields (PNFs) that are compact and easy to optimize and enjoy the interpretability and controllability of signal processing methods. We provided a theoretical framework to analyse the output of a PNF and design a Fourier PNF, whose outputs can be decomposed in a fine-grained manner in the frequency domain with user-specified lower and upper limits on the frequencies. We demonstrated that PNFs matches state-of-the-art performance for signal representation tasks. We then demonstrated the use of PNF's subband decomposition in the settings of texture transfer and scale-space representations. As future work, the ability to generalize our representation to represent multiple higher-dimensional signals (such as multiple images) can enable applications in recognition and generation, where one can leverage our decomposeable architecture to impose a prior or regularize specific subbands to improve generalization.

**Acknowledgement** This research was supported in part by the Pioneer Centre for AI, DNRF grant number P1. Guandao's PhD was supported in part by research gifts from Google, Intel, and Magic Leap. Experiments are supported in part by Google Cloud Platform and GPUs donated by NVIDIA.

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
