# Polynomial Neural Fields
# for Subband Decomposition and Manipulation
# - Supplementary Materials -

**Guandao Yang**[*]
Cornell University

**Sagie Benaim**[*]
University of Copenhagen

**Varun Jampani**
Google Research

**Kyle Genova**
Google Research

**Jonathan T. Barron**
Google Research

**Thomas Funkhouser**
Google Research

**Bharath Hariharan**
Cornell University

**Serge Belongie**
University of Copenhagen

## Contents

---

[*]Equal contribution. Part of this work was done while Guandao was a student researcher at Google.

36th Conference on Neural Information Processing Systems (NeurIPS 2022).

# 1   Theory

In this section, we will provide a detail derivation of the theories we used for building and analyzing PNF. We first study the definition of PNF in Sec. 1.1. Then we prove the properties that PNF is linear sums of basis for different basis in Sec. 1.2 and Sec. 1.3. In Sec. 1.4, we studies how to organize subset of basis, or subbands, in a controllable way to produce subband-limited PNFs. Finally, we show several different instantiation of PNFs in Sec. 1.5.

## 1.1   Definition

**Definition 1.1** (PNF). *Let $\mathcal{B}$ be a basis for the vector space of functions for $\mathbb{R}^n \to \mathbb{R}$. A Polynomial neural field of basis $\mathcal{B}$ is a neural network $f = g_L \circ \cdots \circ g_1 \circ \gamma$, where $\forall i, g_i$ are finite degree multivariate polynomials, and $\gamma : \mathbb{R}^n \to \mathbb{R}^d$ is a $d$-dimensional feature encoding using basis $\mathcal{B}$: $\gamma(x) = [\gamma_1(x), \ldots, \gamma_d(x)]^d, \gamma_i \in \mathcal{B}, \forall i.$*

We will show how this definition has included both $\Pi$-Net [5], MFN [8], and BACON [13].

**$\Pi$-Net.** Here we will set basis $\mathcal{B} = \{x^n\}_{n \geq 0}$. Then $g_i$ can be set according to different factorization mentioned in Section 3.1 and 3.2.

**MFN.** [8] studied two types of MFNs - Fourier and Gabor MFN. For Fourier MFN that that takes $\mathbb{R}^d$ as input, we will set basis to be $\mathcal{B} = \{\sin(\omega^T x)\}_{\omega \in \mathbb{R}^d}$. Assume there are $L_{mfn}$ layers of the multiplicative filter networks and each layer has hidden dimension of $h$. We will set $L = L_{mfn} + 2$ and define $g_i, \gamma$ in the following manner:

$$\gamma \in \mathbb{R}^d \to \mathbb{R}^{L_{mfn}h}, \tag{1}$$

$$g_1(\gamma(x)) = [M_1\gamma(x), \gamma(x)] \in \mathbb{R}^{(L_{mfn}+1)h} \tag{2}$$

$$g_i([z, \gamma(x)]) = [(W_i z + b_i) \odot (M_i \gamma(x)), \gamma(x)], \quad \forall\, 2 \leq i \leq 2 + L_{mfn} \tag{3}$$

$$g_{L_{mfn}+2}([z, \gamma(x)]) = W_{out} x + b_{out}, \tag{4}$$

where $M_i \in \mathbb{R}^{L_{mfn}h \times h}$ selects the $(i-1)h$ to $ih$ basis by setting each row of $M_i$ to be an one-hot vector and only the $((i-1)h)^{th}$ to $(ih)^{th}$ columes are non-zeros. Similarly, for Gabor MFN, we use the same definition of $g$, but switch $\gamma$ to sample from $\mathcal{B} = \{\exp(\gamma \|x - \mu\|^2) \sin(\omega^T x)\}_{(\gamma, \mu, \omega \in \mathbb{R}^d)}$.

**BACON.** The way to instantiate BACON will be similar to MFN. Basically each intermediate output layer of BACON is a MFN with $\gamma$ to be sampled from specific subbands.

The definition of PNF is very general such that it not only include prior works but also allow potential design of new architectures with different network topology and differnet basis. Our design of Fourier PNF will leverage fourier basis with a modified network architecture. We will introduce several more variants of the PNFs with different architectures and basis choices.

## 1.2   Basis function

**Definition 1.2** (Span of Basis is Closed Under Multiplication). *We call a basis $\mathcal{B}$'s span closed under multiplication if : $\forall b_1, b_2 \in \mathcal{B}, b_1(x)b_2(x) = \sum_{i \in I} a_i b_i(x), |I| < \infty.$*

Note that this is the same requirement as Definition 1 in the appendix of MFN paper [9]. We will extend the analysis of MFN in several ways. First, we will show that several commonly used basis functions satisfies Definition 1.2.

**Lemma 1** (Fourier Basis). *Assume the Fourier basis of functions $\mathbb{R}^d \to \mathbb{R}$ takes the form of $\mathcal{B}_{Fourier} = \{b_\omega = \exp(i\omega^T x) | \omega \in \mathbb{R}^d\}$. Then $\mathcal{B}_{Fourier}$'s span is closed under multiplication.*

*Proof.* It's enough to show that the multiplication of two Fourier basis function is still a Fourier basis funciton: $\exp(i\omega_1^T x)\exp(i\omega_2^T x) = \exp(i(\omega_1 + \omega_2)^T x) \in \mathcal{B}_{Fourier}$. □

**Lemma 2** (RBF). *Assume the Radial basis functions for real-value functions $\mathbb{R}^d \to \mathbb{R}$ takes the form of $\mathcal{B}_{RBF} = \{b_{\gamma,\mu} = \exp(-\frac{1}{2}\gamma \|x-\mu\|^2)$. Then $\mathcal{B}_{RBF}$'s span is closed under multiplication.*

*Proof.* Similar to Fourier Basis, we will show that the multiplication of two RBF functions is still in $\mathcal{B}_{RBF}$. This is shown in Equation 24 of Supplementary of MFN [9]:

$$\exp\left(-\frac{1}{2}\gamma_1 \|x-\mu_1\|^2\right) \exp\left(-\frac{1}{2}\gamma_2 \|x-\mu_2\|^2\right) \tag{5}$$

$$= \exp\left(\frac{-\gamma_1\gamma_2 \|\mu_1-\mu_2\|}{2(\gamma_1+\gamma_2)}\right) \exp\left(-\frac{1}{2}(\gamma_1+\gamma_2)\left\|x-\frac{\gamma_1\mu_1+\gamma_2\mu_2}{\gamma_1+\gamma_2}\right\|^2\right) \tag{6}$$

$$= c(\gamma_1,\gamma_2,\mu_1,\mu_2) \exp\left(-\frac{1}{2}\gamma' \|x-\mu'\|^2\right), \tag{7}$$

where $c(\gamma_1,\gamma_2,\mu_1,\mu_2) = \exp\left(\frac{-\gamma_1\gamma_2\|\mu_1-\mu_2\|}{2(\gamma_1+\gamma_2)}\right)$, $\mu' = \frac{\gamma_1\mu_1+\gamma_2\mu_2}{\gamma_1+\gamma_2}$, and $\gamma' = \gamma_1 + \gamma_2$. $\square$

**Lemma 3** (Gabor Basis). *Assume the Gabor basis of functions $\mathbb{R}^d \to \mathbb{R}$ takes the form of $\mathcal{B}_{Gabor} = \{b_{\gamma,\mu,\omega} = \exp(-\frac{1}{2}\gamma \|x-\mu\|^2)\exp(i\omega^T x)|\omega \in \mathbb{R}^d, \mu \in \mathbb{R}^d, \gamma \geq 0\}$. Then $\mathcal{B}_{Gabor}$'s span is closed under multiplication.*

*Proof.* Using Eq. (7), we can compute the the multiplication of two Gabor basis functions:

$$b_{\gamma_1,\mu_1,\omega_1} b_{\gamma_2,\mu_2,\omega_2} = c(\gamma_1,\gamma_2,\mu_1,\mu_2) \exp\left(-\frac{1}{2}\gamma' \|x-\mu'\|^2\right) \exp(i\omega'^T x) \propto b_{\gamma',\mu',\omega'} \in \mathcal{B}_{Gabor}, \tag{8}$$

where $\omega' = \omega_1 + \omega_2$. The output of the multiplication is still a Gabor. $\square$

The following Lemma will show that Definition 1.2 can be extended to analyzing functions from different domain. We will show that for complex-value function that maps from a sphere, there is a basis function that satisfies Definition 1.2.

**Lemma 4** (Spherical Harmonics). *We will consider the basis function for real functions that takes spherical coordinate (i.e. $S^2 \to \mathbb{C}$ where $S^2 = \{(\theta,\phi)|0 \leq \theta \leq \pi, 0 \leq \phi \leq 2\pi\}$. Moreover, we will consider Laplace's spherical harmonics as basis:*

$$\mathcal{B}_{SH} = \{Y_l^m(\theta,\phi) = e^{im\phi} P_l^m(\cos(\theta))|0 \leq l, l \in \mathbb{Z}, -l \leq m \leq l, m \in \mathbb{Z}\}, \tag{9}$$

*where $P_l^m : [-1,1] \to \mathbb{R}$ is an associated Legendre polynomial. $\mathcal{B}_{SH}$ satisfies Definition 1.2.*

*Proof.* From the multiplication rule of Spherical Harmonics [4], we have:

$$Y_{l_1}^{m_1}(\theta,\phi) Y_{l_2}^{m_2}(\theta,\phi) \propto \sum_{l=0}^{\infty} \sum_{m=-c}^{c} (-1)^m \sqrt{2l+1} \begin{pmatrix} l_1 & l_2 & l \\ m_1 & m_2 & -m \end{pmatrix} \begin{pmatrix} l_1 & l_2 & l \\ 0 & 0 & 0 \end{pmatrix} Y_l^m(\theta,\phi), \tag{10}$$

where $\begin{pmatrix} j_1 & j_2 & j_3 \\ m_1 & m_2 & m_3 \end{pmatrix}$ denotes the $3j$-syombols. Now we need to show that the infinite sum contains only finite number of non-zero terms. By the selection rules of the $3j$-symbols [11], we know that $\begin{pmatrix} j_1 & j_2 & j_3 \\ m_1 & m_2 & m_3 \end{pmatrix}$ is zero if any of the following rules is not satisfies: 1) $|j_1 - j_2| \leq j_3 \leq j_1 + j_2$; and 2) $m_1 + m_2 + m_3 = 0$. This implies that for all terms $l \geq l_1 + l_2$, and $m \neq -(m_1 + m_2)$, the term $\begin{pmatrix} l_1 & l_2 & l \\ m_1 & m_2 & -m \end{pmatrix} = 0$. With this said, the Eq. (10) can be written as a finite sum:

$$Y_{l_1}^{m_1}(\theta,\phi) Y_{l_2}^{m_2}(\theta,\phi) \propto \sum_{l=|l_1-l_2|}^{l_1+l_2} \begin{pmatrix} l_1 & l_2 & l \\ m_1 & m_2 & -(m_1+m_2) \end{pmatrix} \begin{pmatrix} l_1 & l_2 & l \\ 0 & 0 & 0 \end{pmatrix} Y_l^m(\theta,\phi). \tag{11}$$

$\square$

There are several rules we can used to generate more basis that satisfies the properties. Here we will show one of them:

**Theorem 5** (Basis-multiplication.). *If $\mathcal{B}_1$ and $\mathcal{B}_2$ be the basis of $\mathbb{R}^d \to \mathbb{R}$ and satisfy Definition 1.2, then $\mathcal{B}_{mult} = \{b_1 b_2 | b_1 \in \mathcal{B}_1, b_2 \in \mathcal{B}_2\}$ also satisfy Definition 1.2.*

*Proof.* Let $b_1 b_2$ and $b_3 b_4$ from $\mathcal{B}_{mult}$. Assume $b_i = \sum_n a_{i,n} b_{k_{i,n}}$, where $i \in \{1, 2, 3, 4\}$, $k_{i,n}$ is an index into $\mathcal{B}_1$ if $i \in \{1, 2\}$ and $\mathcal{B}_2$ if $i \in \{3, 4\}$, and $a_{i,n}$ are coefficients. Then we have

$$(b_1 b_2)(b_3 b_4) = \left( \sum_n a_{1,n} b_{k_{1,n}} \right) \left( \sum_n a_{2,n} b_{k_{2,n}} \right) \left( \sum_n a_{3,n} b_{k_{3,n}} \right) \left( \sum_n a_{4,n} b_{k_{4,n}} \right) \quad (12)$$

$$= \left( \sum_{n,n'} a_{1,n} a_{3,n'} b_{k_{1,n}} b_{k_{3,n'}} \right) \left( \sum_{n,n'} a_{2,n} a_{4,n'} b_{k_{2,n}} b_{k_{4,n'}} \right). \quad (13)$$

Since both $b_{k_{1,n}}$ and $b_{k_{3,n'}}$ are in $\mathcal{B}_1$ and $\mathcal{B}_1$ satisfies Definition 1.2, so we can assume the following (and similarly logics can be applied to $b_{k_{2,n}}$ and $b_{k_{4,n'}}$ for $\mathcal{B}_2$):

$$b_{k_{1,n}} b_{k_{3,n'}} = \sum_m c_{1,n,n',m} b_{l_{1,n,n',m}}, \quad b_{l_{1,n,n',m}} \in \mathcal{B}_1 \quad (14)$$

$$b_{k_{2,n}} b_{k_{4,n'}} = \sum_m d_{n,n',m} b_{l_{2,n,n',m}}, \quad b_{l_{2,n,n',m}} \in \mathcal{B}_2. \quad (15)$$

Plugging abovementioned equations into Eq. (13) we get

$$\sum_{n,n'} a_{1,n} a_{3,n'} b_{k_{1,n}} b_{k_{3,n'}} = \sum_{n,n'} a_{1,n} a_{3,n'} \left( \sum_m c_{1,n,n',m} b_{l_{1,n,n',m}} \right) \quad (16)$$

$$= \sum_{n,n',m} a_{1,n} a_{3,n'} c_{1,n,n',m} b_{l_{1,n,n',m}} = \sum_{n,n',m} \tilde{c}_{1,n,n',m} b_{l_{1,n,n',m}}, \quad (17)$$

for $\tilde{c}_{1,n,n',m} = a_{1,n} a_{3,n'} c_{1,n,n',m}$. Similarly, we have

$$\sum_{n,n'} a_{2,n} a_{4,n'} b_{k_{2,n}} b_{k_{4,n'}} = \sum_{n,n'} a_{2,n} a_{4,n'} \left( \sum_m c_{2,n,n',m} b_{l_{2,n,n',m}} \right) \quad (18)$$

$$= \sum_{n,n',m} a_{2,n} a_{4,n'} c_{2,n,n',m} b_{l_{2,n,n',m}} = \sum_{n,n',m} \tilde{c}_{2,n,n',m} b_{l_{2,n,n',m}}, \quad (19)$$

for $\tilde{c}_{2,n,n',m} = a_{2,n} a_{4,n'} c_{2,n,n',m}$. Finally we can put these two together:

$$(b_1 b_2)(b_3 b_4) = \left( \sum_{n,n',m} \tilde{c}_{1,n,n',m} b_{l_{1,n,n',m}} \right) \left( \sum_{n,n',m} \tilde{c}_{2,n,n',m} b_{l_{2,n,n',m}} \right) \quad (20)$$

$$= \sum_{n,n',m,n'',n''',m'} \tilde{c}_{1,n,n',m} \tilde{c}_{2,n'',n''',m'} b_{l_{1,n,n',m}} b_{l_{2,n'',n''',m'}}, \quad (21)$$

which is a linear combination of basis $\mathcal{B}_{mult}$ since $b_{l_{1,n,n',m}} \in \mathcal{B}_1$ and $b_{l_{1,n'',n''',m'}} \in \mathcal{B}_2$. □

## 1.3 PNF as Linear Sum of Basis

In this section, we will show that PNF can be expressed as linear sum of basis if the basis function satisfies Definition 1.2.

**Lemma 6** (Power-product of Basis). *If $\mathcal{B}$ satisfies Definition 1.2, then power-products of the form $\prod_{n=1}^N b_n^{\alpha_n}$, $0 \le \alpha_n < \infty$ is a linear sum of the basis $\mathcal{B}$.*

*Proof.* We will show by induction on the degree of the power-product: $d = \sum_n \alpha_n$.

*Base cases.* If $d = 0$, then $\prod_{n=1}^N b_n^{\alpha_n} = 1 \in \mathcal{B}$ since $\mathcal{B}$ is a basis function of $\mathbb{R}^d \to \mathbb{R}$.

*Inductive case.* The inductive hypothesis is: assume that for some degree $d$ such that $d \geq 0$, all finite power-product of degree $d$ are linear sums of basis $\mathcal{B}$. We want to show that all finite power-product of degree $d + 1$ is also linear sums of basis $\mathcal{B}$. Condider a power-product of degree $d + 1$ in the form of $\prod_{n=1}^{N} b_n^{\alpha_n}$ and assume without lost of generality that $\alpha_1 \geq 1$. Then we have

$$\prod_{n=1}^{N} b_n^{\alpha_n} = b_1 \left( b_1^{(\alpha_n - 1)} \prod_{n=2}^{N} b_n^{\alpha_n} \right) = b_1 p, \tag{22}$$

where $p = b_1^{(\alpha_n - 1)} \prod_{n=2}^{N} b_n^{\alpha_n}$. It's easy to see that the degree of $p$ is $d$, so $p$ can be written as a linear sum of $\mathcal{B}$: $\sum_m a_m b_m$. With that we have:

$$\prod_{n=1}^{N} b_n^{\alpha_n} = b_1 \sum_m a_m b_m = \sum_m a_m b_1 b_m, \tag{23}$$

where $b_i \in \mathcal{B}$. Since $\mathcal{B}$ satisfies Definition 1.2, so $b_1 b_m$ can be written as a linear sum of basis $\mathcal{B}$: $\sum_j c_j b_j$. Putting this in Eq. (23) gives a linear sum of basis $\mathcal{B}$. $\square$

**Theorem 7.** *Let $F$ be a PNF with basis $\mathcal{B}$. $\forall b_1, b_2 \in \mathcal{B}, b_1(x) b_2(x) = \sum_{i \in I} a_i b_i(x), |I| < \infty$, then the output of $F$ is a finite linear sum of the basis functions from $\mathcal{B}$.*

*Proof.* Let $F = g_L \circ \cdots \circ g_1 \circ \gamma$, with $g_i$ being a multivariate polynomial. Let $z_0 = \gamma$. Let $z_i = g_i \circ \cdots \circ g_1 \circ \gamma$ for $i \geq 1$. Let $z_i[j]$ be the $j^{th}$ dimensional value of $z_i$. We will show by induction that for all $i$, $z_i[j]$ is a linear sum of basis functions from $\mathcal{B}$ for all $j$.

*Base case :* $i = 0$. $z_i[j] = \gamma_j(x) \in \mathcal{B}$ by definition of $\gamma$ in Definition 1.1.

*Inductive case.* The inductive hypothesis is: for $k \geq 1$, if $z_k[j]$ is linear sum of $\mathcal{B}$ for all $j$, then $z_{k+1}[l]$ is linear sums of $\mathcal{B}$ for all $l$.

By definition of $z$, we know that $z_{k+1} = g_k(z_k)$, where $g_k$ is a multivariate polynomial of finite degree $d$. With that said, we can assume $z_{k+1}[l] \in \mathbb{R}$ is a linear sum of power-product terms in the form of $\prod_j z_k[i]^{\alpha_{lj}}$, where $\alpha_{lj} \geq 0$ and $\sum_l \alpha_l \leq d$. It's sufficient to show that each of this term is linear sum of the basis function $\mathcal{B}$.

By the inductive hypothesis, we can assume that $z_k[j] = \sum_n \beta_{jn} b_{jn}$, with $b_{jn} \in \mathcal{B}$. Then we have:

$$\prod_j z_k[i]^{\alpha_{lj}} = \prod_j \left( \sum_n \beta_{jn} b_{jn} \right)^{\alpha_{lj}} = \sum_m a_m \prod_n b_n^{\hat{\alpha}_{nm}}, \tag{24}$$

for some $a_m \in \mathbb{R}$, and $\hat{\alpha}_{nm} \geq 0$ and $\sum_m \hat{\alpha}_{nm} \leq d$. By Theorem 6, the terms $\prod_n b_n^{\hat{\alpha}_{nm}}$ are linear sums of $\mathcal{B}$. Then Eq. (24) is linear sums of linear sums of $\mathcal{B}$, which will be linear sums of $\mathcal{B}$. $\square$

## 1.4 Controllable Sets of Subbands

In this section, we will develop theories to build controllable sets of subbands and use that to design PNFs. In this section, we will use definition that a subband is a subset of the basis function.

**Definition 1.3** (Subband). *A subband $S$ is a subset of $\mathcal{B}$.*

**Definition 1.4** (Subband limited PNF). *A PNF $F$ of basis $\mathcal{B}$ is limited by subband $S \subset \mathcal{B}$ if each dimension of the output (i.e. $F_i$) is in the span of $S$.*

One naive way to construct a subband limited PNF is to restrict $\gamma$ to take only basis functions from the subband: $\gamma_i \in S$, and then restricted no multiplication in the layers $g_i$ (since multiplication is the only operation that can change the composition of the basis functions used in the linear sum).

But this will simply create a very shallow network as the composition of linear layers amounts to only one linear layer. As a result, we will need a very wide network in order to achieve expressivity. This means the number of basis function we used will grow linear with the number of network parameters. When it requires exponential number of basis functions to approximate a signal well, then we are not capable of achieving it compactly, throwing away a key virtual of neural fields.

In order to achieve compactness, we want to be able to generate a lots of basis, all of with within the subband of interest, without instantiating a lot of network parameters. One way to achieve this is to use function composition with non-linearity. In our case, since the functions are restricted to be polynomials, then the only non-linearity we can use is multiplication. But multiplication can potentially create basis functions outside of the subband as Definition 1.2 does not restrict the linear sum to be within certain subband. As a result, we need to study how multiplication transform the subband. More specifically, we will first define a set of subbands, within which multiplication in function space can translate in someway to an operation of subbands:

**Definition 1.5** (PNF-controllable Set of Subbands). $\mathcal{S} = \{S_\theta | S_\theta \subset \mathcal{B}\}_\theta$ *is a PNF-Controllable Set of Subbands for basis* $\mathcal{B}$ *if (1)* $S_{\theta_1} \cup S_{\theta_2} \in \mathcal{S}$ *and (2) there exists a binary function* $\otimes : \mathcal{S} \times \mathcal{S} \to \mathcal{S}$ *such that if* $b_1 \in S_{\theta_1}, b_2 \in S_{\theta_2} \implies b_1 b_2 = \sum_n a_n b_n, b_n \in S_{\theta_1} \otimes S_{\theta_2}$ *for some coefficients* $a_n \in \mathbb{R}$ *(i.e.* $b_1 b_2$ *is in the span of* $S_{\theta_1} \otimes S_{\theta_2}$*).*

**Theorem 8.** *Let* $\mathcal{S}$ *be a PNF-controllable set of subbands of basis* $\mathcal{B}$ *with its corresponding binary function* $\otimes$. *Suppose* $F$ *and* $G$ *are polynomial neural fields of basis* $\mathcal{B}$ *that maps* $\mathbb{R}^n$ *to* $\mathbb{R}^{m_1}$ *and* $\mathbb{R}^{m_2}$ *respectively. Furthermore, suppose* $F$ *and* $G$ *are subband limited by* $S_1 \in \mathcal{S}$ *and* $S_2 \in \mathcal{S}$. *Then we have the following:*

1. $A(x) = \mathbf{w}_1^T F(x) + \mathbf{w}_2 G(x)$ *is a PNF of* $\mathcal{B}$ *limited by subband* $S_1 \cup S_2$ *with* $\mathbf{w}_1 \in \mathbb{R}^{m_1}$ *and* $\mathbf{w}_2 \in \mathbb{R}^{m_2}$ [2]*; and*

2. $M(x) = F(x)^T W G(x)$ *is a PNF of* $\mathcal{B}$ *limited by subband* $S_1 \otimes S_2$ *with* $W \in \mathbb{R}^{m_1 \times m_2}$.

*Proof.* By Definition 1.4, we can assume that

$$\forall 1 \le i \le m_1, F_i(x) = \sum_j a_{f,j} b_{f,j}(x), \quad b_{f,j} \in S_1 \tag{25}$$

$$\forall 1 \le i \le m_2, G_i(x) = \sum_j a_{g,j} b_{g,j}(x), \quad b_{g,j} \in S_2. \tag{26}$$

Then we can compute $A(x)$:

$$A(x) = \sum_{i=1}^{m_1} \mathbf{w}_1[i] F_i(x) + \sum_{i=1}^{m_2} \mathbf{w}_2[i] G_i(x) \tag{27}$$

$$= \sum_{i=1}^{m_1} \mathbf{w}_1[i] \left( \sum_j a_{f,j} b_{f,j}(x) \right) + \sum_{i=1}^{m_2} \mathbf{w}_2[i] \left( \sum_j a_{g,j} b_{g,j}(x) \right) \tag{28}$$

$$= \underbrace{\left( \sum_{i=1}^{m_1} \sum_j (\mathbf{w}_1[i] a_{f,j}) b_{f,j}(x) \right)}_{\text{Subband limited by } S_1} + \underbrace{\left( \sum_{i=1}^{m_2} \sum_j (\mathbf{w}_2[i] a_{g,j}) b_{g,j}(x) \right)}_{\text{Subband limited by } S_2}. \tag{29}$$

It's easy to see that Eq. (29) is subband limited by $S_1 \cup S_2$.

---

[2] In the main text, we show a $o$-dimensional version: $W_1 F(x) + W_2 G(x)$, where $W_1 \in \mathbb{R}^{m_1 \times o}$ and $W_2 \in \mathbb{R}^{m_2 \times o}$. The $o$-dimensional version can be seen as an easy extension of this single dimensional version as we can view $W_1 = [\mathbf{w}_{11}^T; \ldots; \mathbf{w}_{1o}^T]$ where $\mathbf{w}_{1i} \in \mathbb{R}^{m_1}$ and similarly for $W_2$.

Similarly, we can apply the computation to $M(x)$:

$$M(x) = \sum_{i,j \geq 1}^{i,j \leq m_1, m_2} W[i,j] F_i(x) G_j(x) \tag{30}$$

$$= \sum_{i,j \geq 1}^{i,j \leq m_1, m_2} W[i,j] \left( \sum_k a_{f,k} b_{f,k}(x) \right) \left( \sum_k a_{g,k} b_{g,k}(x) \right) \tag{31}$$

$$= \sum_{i,j \geq 1}^{i,j \leq m_1, m_2} W[i,j] \left( \sum_{k,k'} a_{f,k} a_{g,k'} b_{f,k}(x) b_{g,k'}(x) \right) \tag{32}$$

$$= \sum_{i,j \geq 1, k, k'}^{i,j \leq m_1, m_2} (W[i,j] a_{f,k} a_{g,k'}) b_{f,k}(x) b_{g,k'}(x). \tag{33}$$

By Definition 1.5, we can assume

$$b_{f,k}(x) b_{g,k'}(x) = \sum_l c_{k,k',l} b_{k,k',l}(x), \quad b_{k,k',l}(x) \in S_1 \otimes S_2. \tag{34}$$

Putting it into Eq. (33), we have:

$$M(x) = \sum_{i,j \geq 1, k, k'}^{i,j \leq m_1, m_2} (W[i,j] a_{f,k} a_{g,k'}) \sum_l c_{k,k',l} b_{k,k',l}(x) \tag{35}$$

$$= \sum_{i,j \geq 1, k, k', l}^{i,j \leq m_1, m_2} (W[i,j] a_{f,k} a_{g,k'} c_{k,k',l}) b_{k,k',l}(x), \tag{36}$$

which is subband limited by $S_1 \otimes S_2$ since $b_{k,k',l} \in S_1 \otimes S_2$. □

With this theorem, we are able to leverage the PNF-controllable sets of subbands to compose PNFs with multiplication and additions in a band-limited ways. For example, if we want to construct a PNF that's band-limited by $S$, we can do it in the following steps:

1. Identify a PNF-controllable set of subbands that contains $S$, let that be $\mathcal{S}$ with the associated operation to be $\otimes$.

2. Factorize $S$ into a series of subband by the associated operation : $S = S_1 \otimes S_2 \otimes \cdots \otimes S_n$.

3. For each $S_i$, we can creates a shallow PNF with $F_i = g^{(i)} \circ \gamma^{(i)}$, where $\gamma_j^{(i)} \in S_i$ for all $j$ and $g^{(i)}(z) = Wz$ is a linear layer (without bias).

4. Composed these layers together using rule-2 of Theorem 8.

One can see that the abovementioned way to construct PNF has the ability to create exponential number of basis functions, all of which within $S$. Following is the intuitive reason why that's the case. Suppose each neuron $F_i$ is a linear sum of $d$ different basis functions (i.e. the set of basis chosen for $F_i$ is different from those of $F_j$ if $i \neq j$). Furthermore, suppose that Definition 1.2 creates different sets of basis in the right-hand-side of the multiplication. then every time we apply rule-2 of Theorem 8 on $d$-dimensional inputs (i.e. assume that we have $d$-numer of $W$ matrix in rule-2 to create an output of $d$-dimensional everytime), we creates $d$-number of different basis for every-single existing cases. As a result, if we apply rule-2 $L$-times, then we will have $d^L$ number of different basis. At the same time, the number of parameters we used is $d^2$ for each $F_i$ and $d^3$ for each multiplication. This means that the number of parameter is $O(Ld^3)$ while the number of basis functions we create in the final linear sum is $O(d^L)$. As a result, such construction can potentially lead to a compact, expressive, and subband-limited PNF.

Note that what's described above is merely an intuitive argument, since many of the conditions might be difficult to hold strictly (e.g. different basis are created for multiplication on the right-hand-sides of Definition 1.2). Empirically, we found that while these conditions are only loosely held true, such construction is still capable of creating a large number of basis functions. This observation aligns with the analysis of MFN-like network can a large number of basis functions, presented in MFN [9] and BACON [13]. In the rest of the subsection, we will show some construction of PNF-controllable set of subbands with various basis functions.

### 1.4.1 Fourier Basis

In this section, we will consider Fourier basis of $\mathbb{R}^d \to \mathbb{R}$ parameterized by $\omega \in \mathbb{R}^d$: $\mathcal{B}_{Fourier} = \{\exp(i\omega^T x)\}_\omega$. A commonly used subband definition (as shown in Equation (1) of the main text) is following:

**Definition 1.6** (Fourier Subband). *The Fourier subband for Fourier basis of $\mathbb{R}^d \to \mathbb{R}$ functions can be defined with a lower band limit $\alpha \in \mathbb{R}^+$, an upper band limit $\beta \geq alpha$, an orientation $\mathbf{d} \in \mathbb{R}^d$ and an angular width $\gamma \in \mathbb{R}^+$:*

$$R_F(\alpha, \beta, \mathbf{d}, \gamma, p) = \left\{ \boldsymbol{\omega} | \alpha \leq \|\boldsymbol{\omega}\|_p \leq \beta, \|\mathbf{d}\| = 1, \boldsymbol{\omega}^T \mathbf{d} \geq \cos(\gamma) \|\boldsymbol{\omega}\|_2 \right\}. \tag{37}$$

With these subbands, we can find PNF-controllable set of subbands in the following format:

**Theorem 9** (Fourier PNF-Controllable Set of Subbands with L2 norm). *For $|\gamma| < \frac{\pi}{4}$, define $\mathcal{S}_{Fourier-L2}(\mathbf{d}, \gamma) = \{R_F(\alpha, \beta, \mathbf{d}, \gamma, 2) | \forall 0 \leq \alpha \leq \beta\}$. Each subband in $\mathcal{S}$ can be parameterized with tuple $(\alpha, \beta)$. $\mathcal{S}_{Fourier-L2}(\mathbf{d}, \gamma)$ is a PNF-Controllable Set of Subbands with the following definition of binary relation $\otimes_{FF}$:*

$$(\alpha_1, \beta_1) \otimes_{FF} (\alpha_2, \beta_2) = (\sqrt{\cos(2\gamma)}(\alpha_1 + \alpha_2), \beta_1 + \beta_2). \tag{38}$$

*Proof.* It's sufficient to show that $\omega_1 \in R(\alpha_1, \beta_1, \mathbf{d}, \gamma, 2)$, and $\omega_2 \in R(\alpha_2, \beta_2, \mathbf{d}, \gamma, 2)$, then $\omega_1 + \omega_2 \in R(\sqrt{\cos(2\gamma)}(\alpha_1 + \alpha_2), \beta_1 + \beta_2, \mathbf{d}, \gamma, 2)$. Let's start with the bound in the norm:

$$\|\omega_1 + \omega_2\| = \sqrt{\|\omega_1\| + \|\omega_2\| + 2\omega_1^T \omega_2} \tag{39}$$

$$= \sqrt{\|\omega_1\|^2 + \|\omega_2\|^2 + 2\|\omega_1\| \|\omega_2\| \cos(\theta)}, \tag{40}$$

where $\theta$ is the radius of the angle between $\omega_1$ and $\omega_2$. Since $\cos(\theta) \leq 1$, it's easy to show that:

$$\|\omega_1 + \omega_2\| = \sqrt{\|\omega_1\|^2 + \|\omega_2\|^2 + 2\|\omega_1\| \|\omega_2\| \cos(\theta)} \tag{41}$$

$$\leq \sqrt{\beta_1^2 + \beta_2^2 + 2\beta_1\beta_2} = \beta_1 + \beta_2. \tag{42}$$

Now we need to show that $\|\omega_1 + \omega_2\| \geq \alpha_1 + \alpha_2$. By Eq. (37), we know that $\omega_1$'s angle with $\mathbf{d}$ is at most $\gamma$. Similarly, $\omega_2$'s angle with $\mathbf{d}$ is also at most $\gamma$. With that said, the largest angle between $\omega_1$ and $\omega_2$ should be less than $2\gamma$. As a result, $\cos(\theta) \geq \cos(2\gamma)$. Note that $|\gamma| \leq \frac{\pi}{4}$, so $\cos(\theta) \geq \cos(2\gamma) \geq \cos(\frac{\pi}{2}) = 0$. With that, we can show:

$$\|\omega_1 + \omega_2\| = \sqrt{\|\omega_1\|^2 + \|\omega_2\|^2 + 2\|\omega_1\| \|\omega_2\| \cos(\theta)} \tag{43}$$

$$\geq \sqrt{\alpha_1^2 + \alpha_2^2 + 2\|\omega_1\| \|\omega_2\| \cos(\theta)} \tag{44}$$

$$\geq \sqrt{\alpha_1^2 + \alpha_2^2 + 2\alpha_1\alpha_2 \cos(2\gamma)} \quad \text{(Possible since } \cos(\theta) \geq \cos(2\gamma) \geq 0) \tag{45}$$

$$= \sqrt{\cos(2\gamma)(\alpha_1^2 + \alpha_2^2 + 2\alpha_1\alpha_2) + (1 - \cos(2\gamma))(\alpha_1^2 + \alpha_2^2)} \tag{46}$$

$$\geq \sqrt{\cos(2\gamma)(\alpha_1^2 + \alpha_2^2 + 2\alpha_1\alpha_2)} \quad \text{Since } \cos(2\gamma) \leq 1 \text{ and } \alpha_1, \alpha_2 \geq 0 \tag{47}$$

$$= \sqrt{\cos(2\gamma)}(\alpha_1 + \alpha_2). \tag{48}$$

Finally, we want to show that $(\omega_1 + \omega_2)^T \mathbf{d} \geq \cos(\gamma) \|\omega_1 + \omega\|$:

$$(\omega_1 + \omega_2)^T \mathbf{d} = \omega_1^T \mathbf{d} + \omega_2^T \mathbf{d} \tag{49}$$

$$\geq \cos(\gamma) \|\omega_1\| + \cos(\gamma) \|\omega_2\| \tag{50}$$

$$= \cos(\gamma)(\|\omega_1\| + \|\omega_2\|) \tag{51}$$

$$\geq \cos(\gamma) \|\omega_1 + \omega_2\| \quad \text{by Triangular inequiality of L2-norm.} \tag{52}$$

$\square$

Intuitively, the smaller the angel $\gamma$ is, the tighter we are able to guarantee the lower-bound compared to $\alpha_1 + \alpha_2$. With this said, if we want to build a subband with lower-bound $l$ and angle $\gamma$, one way to achieve it is to set $\alpha_1 = 0$ and $\alpha_2 = \frac{l}{\sqrt{\cos(2\gamma)}}$. But we found empirically that the network perform better when the lower-band-limit is overlap with the upper-band-limit. We will discuss this more in Sec. 2.

### 1.4.2 Fourier L1

As mentioned in Section 3.3.2 of the main paper, when working in image domain, a preferable way to define the PNF-controllable set of subbands is through L-$\infty$ norm. This is because the L-$\infty$ norm creates subband that can tile the corner of the frequency domain created by the image without going over the Nyquist rate. But to achieve it, we need to restrict each subband within a total-vertical or total-horizontal region. Intuitively, those regions contains vectors whose L-$\infty$ norm is taking absolute value of the same dimension:

**Definition 1.7** (L-$\infty$ dimension consistent region). *We call a region $R_\infty(n, s) \subset \mathbb{R}^d$ has consistent L-$\infty$ dimension of $n$ and sign $s \in \{+1, -1\}$ if:*

*1. $\forall \mathbf{d} \in R, \|\mathbf{d}\|_\infty = |\mathbf{d}[n]|$, where $\mathbf{d}[n]$ denotes the nth value of vector $\mathbf{d}$; and*

*2. $\forall \mathbf{d}_1, \mathbf{d}_2 \in R, \text{sign}(\mathbf{d}_1[n]) = \text{sign}(\mathbf{d}_2[n]) = s$.*

Please refer to Fig. 1 for the four L-$\infty$ dimension consistent region in $\mathbb{R}^2$. In general, for $\mathbb{R}^d$, there are $2d$ number of such regions.

It's easy to see the following property of $R_\infty(n)$:

**Lemma 10.** *If $\mathbf{d} \in R_\infty(n, s)$, then $a\mathbf{d} \in R_\infty(n, s)$ for $a \in \mathbb{R}$ and $a > 0$.*

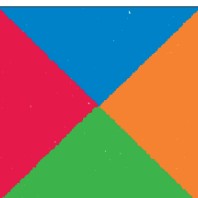

*Proof.*

$$\arg\max_i |(a\mathbf{d})[i]| \arg\max_i |a\mathbf{d}[i]| = \arg\max_i |a||\mathbf{d}[i]| = \arg\max_i |\mathbf{d}[i]| = n. \quad (53)$$

Figure 1: Visualization of the four L-$\infty$ consistent regions in $\mathbb{R}^2$, one color for each.

Since $a > 0$, $\text{sign}(a\mathbf{d}[n]) = \text{sign}(\mathbf{d}[n])$. $\square$

With this, we are able to prove a similar version of Theorem 9 for L-$\infty$ norm but restricted each set of subband to be within only a region with consistent L-$\infty$ dimension:

**Theorem 11** (Fourier PNF-Controllable Set of Subbands with L-$\infty$ norm). *Let $R_\infty(n, s)$ be a region of consistent L-$\infty$ dimension of $n$ defined for $\mathbb{R}^d$. Define the set of subbands as following:*

$$\mathcal{S}_{FF-L\infty}(\mathbf{d}, \gamma, n) = \{R_F(\alpha, \beta, \mathbf{d}, \gamma, \infty) | \forall 0 \leq \alpha \leq \beta, R_F(\alpha, \beta, \mathbf{d}, \gamma, \infty) \subset R_\infty(n, s)\}. \quad (54)$$

*Each subband in $\mathcal{S}_{FF-L\infty}$ can be parameterized with tuple $(\alpha, \beta)$. $\mathcal{S}_{FF-L\infty}(\mathbf{d}, \gamma)$ is a PNF-Controllable Set of Subbands with the following definition of binary relation $\otimes_{FF}$:*

$$(\alpha_1, \beta_1) \otimes_{FF\infty} (\alpha_2, \beta_2) = (\alpha_1 + \alpha_2, \beta_1 + \beta_2). \quad (55)$$

*Proof.* Similar to the proof in Theorem 9, we will show that if $\omega_1 \in R_F(\alpha_1, \beta_1, \mathbf{d}, \gamma, \infty) = R_1 \in \mathcal{S}_{FF-\infty}$ and $\omega_2 \in R_F(\alpha_2, \beta_2, \mathbf{d}, \gamma, \infty) = R_2 \in \mathcal{S}_{FF-\infty}$, then we will have $\omega_1 + \omega_2 \in R_F(\alpha_1 + \alpha_2, \beta_1 + \beta_2, \mathbf{d}, \gamma, \infty) \in \mathcal{S}_{FF-\infty}$.

First, we show that $\omega_1 + \omega_2$ is still within $R_\infty(n)$. As shown in the proof for Theorem 9, we have $(\omega_1 + \omega_2)^T \mathbf{d} \geq \cos(\gamma) \|\omega_1 + \omega_2\|$. This means that $\omega_1 + \omega_2 \in R_\infty(n)$ by Theorem 10.

Since $R_1, R_2$, and $\omega_1 + \omega_2$ are both in $\mathcal{S}_{FF-\infty}$, we will use this property to derive the upper bound:

$$\|\omega_1 + \omega_2\|_\infty = |(\omega_1 + \omega_2)[n]| \leq |\omega_1[n]| + |\omega_2[n]| = \|\omega_1\|_\infty + \|\omega_2\|_\infty = \beta_1 + \beta_2. \quad (56)$$

If $s > 0$, then we know $\omega_1[n], \omega_2[n] > 0$ and $(\omega_1 + \omega_2)[n] > 0$. As a result, $\|\omega_{1,2}\|_\infty = \omega_{1,2}[n] \geq \alpha_{1,2}$ respectively. This implies $\|\omega_1 + \omega_2\|_\infty = \omega_1[n] + \omega_2[n] \geq \alpha_1 + \alpha_2$.

Similar arguement can be applied when $s < 0$. $\square$

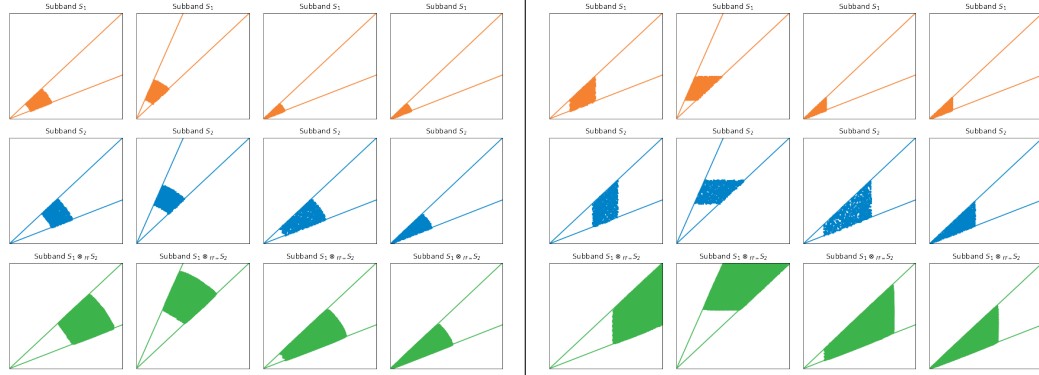

Figure 2: Illustration of how subbands within PNF-Controllable Sets of subbands transform under multiplication. We illustrate subbands defined in **??**. So x-axis is $\omega[1]$ and y-axis is $\omega[2]$. The left four figures shows the operation under $\otimes_{FF}$ (Theorem 9) and the right four figures shows operations under $\otimes_{FF\infty}$ (Theorem 11). The top two rows are $S_1$ and $S_2$, and the last row is $S_1 \otimes S_2$ with $\otimes$ to be the corresponding binary functions defined in the PNF-controllable set of subbands.

Note that this theorem shows that the rectangular tiling mentioned in Equation (3) of the main text is actually operating under a PNF-controllable set of subbands. We also provide an illustration of these two subband binary functions in Fig. 2.

### 1.4.3 RBF

**Theorem 12** (RBF PNF-Controllable Set of Subbands). *Assume the RBF-basis functions for $\mathbb{R}^d \to \mathbb{R}$ in the form $\mathcal{B}_{RBF} = \{\exp(-\frac{1}{2}\gamma \|x - \mu\|)\}$ parameterized by tuple $(\gamma, \mu)$, where $\gamma \in \mathbb{R}$ and $\mu \in \mathbb{R}^d$. Define subband in the following way:*

$$S_{RBF}(M, \gamma) = \{\exp(-\frac{1}{2}\gamma \|x - \mu\|) | \mu \in \mathrm{Cvx}(M)\}, \tag{57}$$

*where $M \subset \mathbb{R}^d$ and $\mathrm{Cvx}(M)$ denotes the convex hull using all vectors of $M$. Then*

$$\mathcal{S}_{RBF} = \{S_{RBF}(M, \gamma) | \forall M \subset \mathbb{R}^d, |M| < \infty, \gamma > 0\} \tag{58}$$

*is a PNF-controllabl Set of Subbands with the following definition of binary relation $\otimes_{RBF}$:*

$$(M_1, \gamma_1) \otimes_{RBF} (M_2, \gamma_2) = (M_1 \cup M_2, \gamma_1 + \gamma_2). \tag{59}$$

*Proof.* (Sketch) This is because by applying Eq. (7) , we have $\mu'$ is a weighted sum of $\mu_1$ and $\mu_2$ with weights normalized to 1. With that said, $\mu'$ is in the convex hull of where $\mu_1$ and $\mu_2$ is sampling from. The operation on $\gamma$ is taken directly from Eq. (7). □

We can easily generalize Theorem 12 to subband definition where $\gamma$ samples from an interval since it only makes sense when $\gamma > 0$. The abovementioned theory suggests that every-time we multiply two RBF basis PNF, we will increase the region of the convex hull of $M$ and increase the $\gamma$ (which is inverse to the scale of the RBF).

### 1.4.4 Gabor

We will show how to combine RBF and Fourier cases of the PNF-controllable set of subbands to create a PNF-controllable set of subband for Gabor basis.

**Theorem 13** (Multiplication Rule of PNF-Controllable Set of Subbands). *Let $\mathcal{B}_1$ and $\mathcal{B}_2$ be two basis for function $\mathbb{R}^d \to \mathbb{R}$ that satisfies Definition 1.2. Let $\mathcal{S}_1, \otimes_1$ and $\mathcal{S}_2, \otimes_2$ be the PNF-controllable set of subbands for $\mathcal{B}_1$ and $\mathcal{B}_2$ respectively. Assume the subbands for $\mathcal{B}_1$ and $\mathcal{B}_2$ are parameterized by $\theta_1 \in \mathbb{R}^n$ and $\theta_2 \in \mathbb{R}^m$ correpsondingly. Define $\mathcal{B}_3 = \{b_1 b_2 | b_1 \in \mathcal{B}_1, b_2 \in \mathcal{B}_2\}$. Define subband*

*as $S_3(\theta_1, \theta_2) = \{b_1 b_2 | b_1 \in S_1(\theta_1), b_2 \in S_2(\theta_2)\}$, where $S_1$ and $S_2$ are subbands for $\mathcal{B}_1$ and $\mathcal{B}_2$ correspondingly. Then following is a PNF-controllable set of subbands for basis $\mathcal{B}_3$:*

$$\mathcal{S}_3 = \{S_3(\theta_1, \theta_2) | S_1(\theta_1) \in \mathcal{S}_1, S_2(\theta_2) \in \mathcal{S}_2\}, \tag{60}$$

*whose corresponding binary function is defined as :*

$$(\theta_{1,a}, \theta_{2,a}) \otimes_3 (\theta_{1,b}, \theta_{2,b}) = (\theta_{1,a} \otimes_1 \theta_{1,b}, \theta_{2,a} \otimes_2 \theta_{2,b}). \tag{61}$$

*Proof.* (Sketch) It's easy to show that $\mathcal{B}_3$ satisfies Definition 1.2 using Theorem 5. Then we can show that multiplication of two basis function in $\mathcal{B}_3$ moves $S_3$ according to $\otimes_3$ by leveraging the definition of $\mathcal{S}_1, \otimes_1$ and $\mathcal{S}_2, \otimes_2$. $\qquad \square$

The PNF-controllable sets of subbands for Gabor basis can be obtained by applying Theorem 13 to combine a Fourier set of subbands (e.g. Theorem 9) and the RBF set of subbands (e.g. Theorem 12).

## 1.5 Different Instantiation of PNFs

In the previous sections, we've shown that 1) PNF allows different architectures, and 2) PNF allows different choices of basis, and 3) PNF can be designed to be subband-limited without losing its compositionality. In this section, we will show several instantiation of the PNFs. Specifically, for each design of the PNF, we will use the following steps:

1. Identify the subband of interests. These subbands should be able to cover all necessary basis functions needed to reconstruct the signal correctly.
2. Organize the subbands according to the PNF-Controllable sets of subbands.
3. For each PNF-Controllable set of subband, create a PNF whose outputs are subband-limited to the corresponding subband of interests in the set.
4. The final PNF is subs of all the previous PNF.

In this section we will use the abovementioned framework to show how PNF can be instantiated in a different forms, using different basis and network architectures. The detailed instantiation of Fourier PNF will be discussed in Sec. 2.

**Gabor PNF.** Similar to MFN [9], we use the same network architecture as Fourier PNF, but changing the basis into a Gabor basis (as shown in Theorem 3). With the Gabor basis, the definition of the subbands requires to include a partition in the spatial domain as shown in Theorem 12. For simplicity, we can set $\gamma \in [0, \infty)$ (i.e not trying to control $\gamma$) and set $M$ in the following way:

$$M = \left\{ s_i \mathbf{e}_i + s_j \mathbf{e}_j | i \neq j, s_{i,j} \in \{0.5, -0.5\}, \mathbf{e}_i[k] = \begin{cases} 1 & \text{if } k = i \\ 0 & \text{otherwise} \end{cases} \right\}. \tag{62}$$

Intuitively, $\mathbf{e}_i$ is an one-hot vector with the $i^{th}$ dimension to be 1. An example of $M$ in 2D is $\{[0.5, 0.5], [0.5, -0.5], [-0.5, 0.5], [-0.5, -0.5]\}$, which includes all the points in the rectangle of $[-0.5, 0.5]^2$. Since all data are sampled withitn $[-0.5, 0.5]^d$, this means the only contorl we want to enforce on $\mu$ is that it should live within the boundary $[-0.5, 0.5]^d$. With this, we are able to create a subband-limited version of Gabor MFN, whose output is a linear sums of the Gabor basis.

**RBF PNF.** Here we will show an interesting way to design an RBF PNF which corresponds to subdividing a rectangular grid and interpolating it with a Gaussian. Here we will use the subband defined by Definition 1.5. We are interesting in modeling the following sets of subbands, one for each level $l$: $S_{RBF}(l) = S_{RBF}(M_l, 2^n \gamma_0)$, where $M_l = \{\sum_{i=1}^{d} \frac{k_i}{2^l} \mathbf{e}_i | k_i \in [-2^{l-1}, 2^{l-1}]\}$ is a set of grid with resolution $2^l$. While $M_l$ contains $2^{2l}$ number of basis, this can be created through function composition very compactly through PNF. First, define $f(x) \in \mathbb{R}^{d^2}$ that each $f(x)[i]$ is an RBF function which takes a corner in $[-0.5, 0.5]^d$ as $\mu$ and $\gamma_0$ as the scale. Then we define the network in the following recursive way:

$$F_0 = f(x), \quad F_{k+1} = (A_{k+1} F_k(x)) \odot (B_{k+1} F_k(x)), \tag{63}$$

where $\odot$ denotes Hardarmard product and $A, B$ are real-value matrices. We can use Theorem 8 and Theorem 12 to show that $F_k$ is subband limited by $S_{RBF}(k)$ and it creates all $\mu$'s that's defined by $M_l$. The output of $F_k$ can be viewed as an Gaussian interpolated version of points $M_l$, where the value of each $M_l$ is given by the network parameters $A$'s and $B$'s.

**Pairwise Gabor PNF.** For the previous definition of Gabor PNF, we are not attempting to control the spatial content of the signal (i.e. there is no ability to pull out a network output with specific $\gamma$ and $M$ region). Fortunately, the PNF allows us to design a netowrk architecture to enable such control for both the spatial RBF part and the Fourier part. The idea is to have a branch of the network to generate RBF PNF $H_{k,l}$ that's subband limited by different $M_k$ at different levels of $2^l\gamma_0$. This is achievable by creating an essemble of network using equation Eq. (63). Similarly, Fourier PNF creates a series of PNFs $G_{n,m}$, each of which is subband limited in different scale (indexed by $n$) and different orientation (indexed by $m$). We want to design the Pairwise Gabor PNF to be the sum of pairwise product of $F_{k,l}$ and $G_{n,m}$:

$$F(x) = \sum_{k,l,n,m} a(k,l,n,m)H_{k,l}(x)G_{n,m}(x), \tag{64}$$

where $a_{k,l,n,m}$ are trainable parameters. It's easy to show that the output of such design is subband limited applying Theorem 13. And this network design allows us to remove either certain spatial area by setting $a_{k,l,\cdot,\cdot} = 0$, or to remove certain frequency content by setting $a_{\cdot,\cdot,n,m} = 0$.

We will show results of some of these designs in the image expressivity experiments (i.e. Fig. 4.

## 2 Implementation Details

As mentioned in Sec. 3.3.3 of the main text, we leverage Theorem 2 to factorize $F$

$$F(\mathbf{x}) = \sum_j F_j(\mathbf{x}), F_j(\mathbf{x}) = G_j(\mathbf{x}, b_j, b_j)W_{jn}Z_{j,n}(\mathbf{x}), \tag{65}$$

$$Z_{j,1}(\mathbf{x}) = G_j(\mathbf{x}, 0, \Delta_1), Z_{j,k}(\mathbf{x}) = G_j(\mathbf{x}, 0, \Delta_k)W_iZ_{j,k-1}(\mathbf{x}), \tag{66}$$

where $G_j(\mathbf{x}, a, b)$ is subband limited in $R_F(a, b, d(\theta_j), \delta, \infty)$ and $\Delta_k = b_k - b_{k-1}$. We instantiate this architecture by setting $G_j(\mathbf{x}, a, b)$ into a linear transform of basis sampled from the subband to be limited:

$$G_j(\mathbf{x}, a, b) = W_i\gamma_j(\mathbf{x}), \gamma_j \in R_F(a, b, d(\theta_j), \delta, \infty)^d, W_i \in \mathbb{R}^{h \times d}, \tag{67}$$

where $h$ and $d$ is the dimension for the output and the feature encoding. We realize the $W_i$ as linear layers with no bias. $\gamma_j \in R_F(a, b, d(\theta_j), \delta, \infty)^d$ are initialized randomly, with $\theta$ and the radius chosen uniformly in $R_F(a, b, d(\theta_j), \delta, \infty)^d$.

For experiments involving 2D images (e.g., image fitting, texture manipulation), the output dimension $h$ for $G_j(\mathbf{x}, b_j, b_j)$ is set to be 3 (RGB output values). The input dimension $d$ for $Z_{j,1}(\mathbf{x}) = G_j(\mathbf{x}, 0, \Delta_1)$ is set to be 2 ($x$ and $y$ coordinates). Otherwise the hidden dimensions are chosen to be 128. For 3D SDF fitting, a hidden dimension of size 100 is chosen, and for NeRF, a hidden dimension of 86 is chosen. The input dimension for 3D SDF fitting and for NeRF is 3 (we follow BACON's setting of modeling irradiance fields). The output dimension is 1 for 3D SDF fitting and 4 for NeRF (RGB and Occupancy values).

**Tiling** For images, the region of interest chosen for tiling is $[-B, B]^2$ where $B$ is band limit, set to be 64, following BACON [13]. Eq. 2 and Eq. 3 describe a potential tiling with no overlapping fans. In practice, for fitting tasks we found it beneficial to use overlapping fans, and so consider the following, modified tilings:

$$\mathcal{T}_{circ} = \{S_{ij} = R_F(b_i, b'_{i+1}, \mathbf{d}(\theta_j), \delta, 2)|b_1 \leq \cdots \leq b_{n-1}, b'_2 \leq \cdots \leq b'_n,$$
$$\theta_j = j\delta, \delta = \frac{\pi}{m}, 1 \leq j \leq 2m\},$$

$$\mathcal{T}_{rect} = \{S_{ij} = R_F(b_i, b'_{i+1}, \mathbf{d}(\theta_j), \delta, \infty)|b_1 \leq \cdots \leq b_{n-1}, b'_2 \leq \cdots \leq b'_n,$$
$$\theta_j = j\delta, 1 \leq j \leq 2m, j \neq m\}.$$

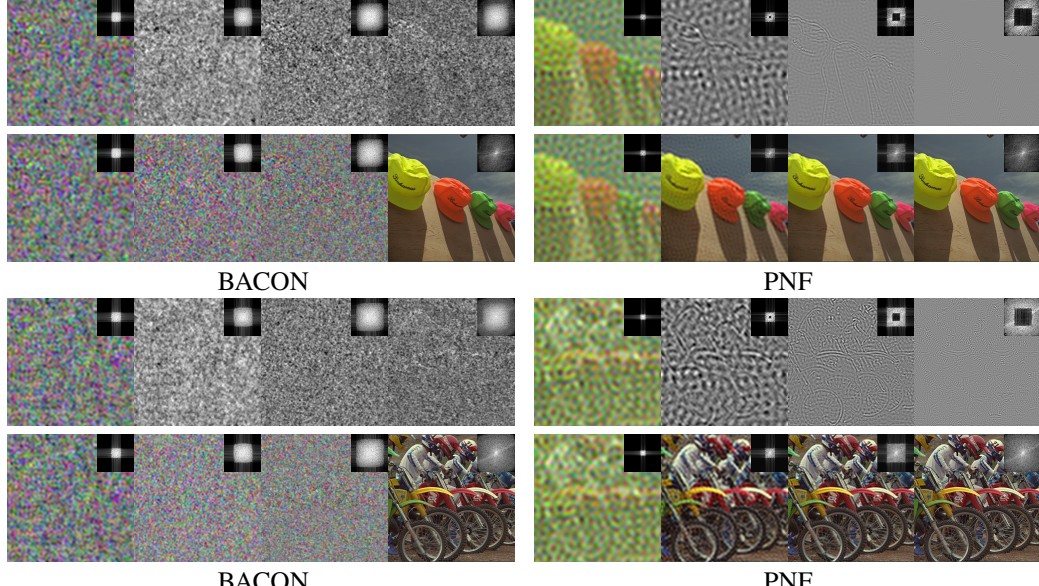

Figure 3: BACON: The bottom row shows the output of each layer which is upper band limited. The top row (columns 2-4) shows the difference between the output of a given layer and the one before it. PNF: The top row shows the output of each layer which is both upper and lower band limited. The bottom row (columns 2-4) shows the addition of the output at a given layer and the one before it.

This allows us to cover the space of frequency basis more compactly and learn a more faitfull fitting of a given signal. In particular, we set $b_1 \leq \cdots \leq b_{n-1}$ to be $0, \frac{1}{16}, \frac{1}{8}, \frac{1}{4}$ and $b'_2 \leq \cdots \leq b'_n$ to be $0, \frac{1}{8}, \frac{1}{8}, \frac{1}{4}, \frac{1}{2}$. $m$ is chosen to be $\frac{1}{8}$, covering the frequency band with 8 orientations. By default we use rectangular tiling for our experiments with PNF.

For experiments that requires higher dimensional inputs (e.g. 3D), we mainly use a generalization of the rectangular tiling. We will first create a tile for each L-$\infty$ dimensional consistent region. In the case of 3D, we will create one PNF-controllable set of subbands for each of the following: $R_\infty(1, 1)$, $R_\infty(2, 1)$, and $R_\infty(3, 1)$. We use the same division of bandwidth within the construction of each of these PNF-controllable subband sets, but scaled it to correpsnding max bandwidth according to different applications. While this already covers all the basis function of interests, we found it improves the performance if we tile the frequency space in an overcomplete way. Specifically, for each pairs of the eight octants in $[-B, B]^3$ where $B$ is the band-limit, we will create three non-overlapping PNF-controllable sets of subbands with the L-$\infty$ norm. One interesting trick we leveraged is that the three non-overlapping PNF-controllabel sets of subband within the same octant can be implemented with one band-limited PNF. With this said, for 3D, we will need one band-limited PNF for each axis and one for each pairs of octants. This leads to an enssemble 7 band-limited PNFs.

## 3 Experiment Details

### 3.1 Expressivity

**Images** For the image fitting task (Sec. 4.1, Tab. 1), we use the DIV2K [1] dataset and downsample images to $256^2$ resolution. For evaluation, we sample the fields at $512$ and compare with the original resolution images. We compare our method against state-of-the-art neural fields of BACON [13], Random Fourier Features [16] and SIREN [15]. Fig. 4 corresponds to Tab.1 of the main paper. We add here SD values corresponding to an averaged over all 25 images in DIV2K [1] dataset. In the paper we report values rectangular tiling. In Fig. 4 we also add the values corresponding to circular tiling. Additional results corresponding to Fig. 2 of the main paper are given in Fig. 3

Figure 4: Image Fitting on the DIV2K dataset. BAC stands for BACON. Rectangular stands for PNF using $\mathcal{T}_{rect}$ tiling, and Circular stands for PNF using $\mathcal{T}_{circ}$ tiling. Gabor is Gabor PNF and PWGabor is the Pairwise Gabor PNF, both are described in Sec. 1.5

| Method | PSNR | SSIM |
|---|---|---|
| RFF | $28.72 \pm 2.88$ | $0.834 \pm 0.053$ |
| SIREN | $29.22 \pm 3.08$ | $0.866 \pm 0.053$ |
| BAC | $28.67 \pm 2.83$ | $0.838 \pm 0.048$ |
| BAC-L | $29.44 \pm 3.03$ | $0.871 \pm 0.047$ |
| BAC-M | $29.44 \pm 3.03$ | $0.871 \pm 0.047$ |
| PNF (Rectangular) | $\mathbf{29.47 \pm 3.08}$ | $0.874 \pm 0.047$ |
| PNF (Circular) | $28.83 \pm 3.04$ | $0.856 \pm 0.052$ |
| PNF (Gabor) | $29.39 \pm 2.97$ | $\mathbf{0.875 \pm 0.047}$ |
| PNF (PWGabor) | $29.22 \pm 2.95$ | $0.872 \pm 0.047$ |

**Neural Radiance Field**    The method devised by NeRF [14] can be used for novel view synthesis. It operates on a dataset of images from different views with known camera parameters. NeRF queries a neural field $M$ for an RGB and occupancy value, given a 3D point on a ray which passes through an image pixel that extends from a camera center. The RGB and occupancy values are then aggregated using standard volumetric rendering pipeline. After training, novel views are rendering by evaluating the relevant rays. For test views, evaluation is performed by measuring the difference between generated views and ground truth views, using SSIM and PSNR measures.

For the choice of neural field $M$, we evaluate our method on the multiscale Blender dataset [2] with images at full ($64\times64$), $1/2$, $1/4$, and $1/8$ resolution. We compare our method to BACON which is state-of-the-art on this task. We use the same training scheme as in BACON for this task. In Fig. 4, we also provide a visual comparison for the drums scene, trained at full ($512\times512$), $1/2$, $1/4$, and $1/8$ resolution.

An Adam optimizer is used for training with $1e6$ training iterations. Learning rate is annealed logarithmically from $1e-3$ to $5e-6$. For BACON, 8 hidden layers are used with 256 hidden features. As mention in Sec. 2, for our network, 4 layers are used with hidden dimension of 86. This results in the total memory which is slightly below that of BACON. Rays for the multiscale Blender dataset are in $[-4, 4]^3$. We follow follow BACON in setting the maximum bandwidth to be 64 cycles per unit interval and in evaluating without the viewing direction as input. We also adapt the hierarchical sampling of NeRF [14]. For a fair comparison to BACON we consider BACON's per-scale supervision using the loss of $\sum_{i,j,k} \|(\mathbf{I}_k(\mathbf{r}_i, \mathbf{t}_j) - \mathbf{I}_{\mathrm{GT},k}(\mathbf{r}_i)\|_2^2$, for $i$, $j$, and $k$ being index rays, ray positions, and dataset scales. Tab. 3 of the main text provides the result averaged over all scenes in the Blender dataset for $1x$ ($64\times64$) resolution and for the average over $1x$, $1/2$, $1/4$, and $1/8$ resolution. Full results are provided in Tab. 1.

**3D Signed Distance Field**    As mentioned in the main text, we evaluate the performance of our method against the Stanford 3D scanning repository[3]. In Tab. 2 of the main text we report the averaged for the scenes of *Armadillo*, *Dragon*, *Lucy*, and *Thai Statue*. Individual per-object scores are provided in Tab. 2 where we consider an additional evaluation metric of normal consistency. Normal consistency (NC), first computes the nearest points using Campfer Distance and then then computes whether the surface normal is within certain threshold. NC is the percentage that lands within the threshold. In Tab. 2, oracle corresponds to the upper bound for the performance, computed by sampling two sets of points form the ground truth and computing the evaluation metrics on them. We compare our method to SIREN [15] and BACON [13]. We train each network to fit a signed distance function (SDF). For BACON and SIREN, 8 hidden layers are used with 256 hidden features. The models are extracted at $512^3$ resolution using marching cubes and evaluated using F-score and Chamfer distance.

---

[3] http://graphics.stanford.edu/data/3Dscanrep/

Table 1: NeRF Fitting for $64^2$ resolution. Full results on 1x ($64^2$), 1/2, 1/4, 1/8 resolutions.

| | BACON [13] | | | | PNF (Ours) | | | |
| --- | --- | --- | --- | --- | --- | --- | --- | --- |
| | 300 epochs | | 500 epochs | | 300 epochs | | 500 epochs | |
| | PSNR | SSIM | PSNR | SSIM | PSNR | SSIM | PSNR | SSIM |
| LEGO 1x | 29.62 | 0.969 | 30.07 | 0.973 | 30.40 | 0.967 | 31.10 | 0.965 |
| LEGO 1/2x | 29.90 | 0.955 | 30.65 | 0.957 | 30.41 | 0.962 | 31.17 | 0.951 |
| LEGO 1/4x | 29.91 | 0.939 | 30.46 | 0.939 | 31.08 | 0.947 | 31.75 | 0.944 |
| LEGO 1/8x | 28.87 | 0.913 | 28.85 | 0.917 | 30.39 | 0.921 | 30.86 | 0.934 |
| Chair 1x | 29.89 | 0.914 | 30.93 | 0.958 | 31.05 | 0.966 | 30.91 | 0.983 |
| Chair 1/2x | 30.97 | 0.898 | 34.93 | 0.968 | 35.00 | 0.949 | 34.88 | 0.975 |
| Chair 1/4x | 30.40 | 0.943 | 37.04 | 0.973 | 37.16 | 0.937 | 37.06 | 0.956 |
| Chair 1/8x | 29.65 | 0.947 | 35.92 | 0.964 | 36.29 | 0.929 | 36.55 | 0.956 |
| Drums 1x | 26.10 | 0.937 | 28.24 | 0.923 | 28.30 | 0.932 | 28.12 | 0.958 |
| Drums 1/2x | 28.38 | 0.942 | 30.20 | 0.956 | 30.05 | 0.939 | 30.16 | 0.960 |
| Drums 1/4x | 28.63 | 0.924 | 31.09 | 0.946 | 31.52 | 0.925 | 31.44 | 0.955 |
| Drums 1/8x | 32.79 | 0.950 | 32.25 | 0.940 | 32.64 | 0.954 | 32.96 | 0.946 |
| Ficus 1x | 25.61 | 0.925 | 28.18 | 0.953 | 29.31 | 0.965 | 30.34 | 0.976 |
| Ficus 1/2x | 31.49 | 0.955 | 33.12 | 0.975 | 30.37 | 0.957 | 30.45 | 0.962 |
| Ficus 1/4x | 35.48 | 0.973 | 35.55 | 0.979 | 30.96 | 0.945 | 30.33 | 0.943 |
| Ficus 1/8x | 37.98 | 0.971 | 38.00 | 0.977 | 30.35 | 0.922 | 28.09 | 0.912 |
| Hotdog 1x | 30.82 | 0.981 | 33.82 | 0.981 | 34.62 | 0.966 | 34.80 | 0.983 |
| Hotdog 1/2x | 29.39 | 0.973 | 34.37 | 0.974 | 34.15 | 0.949 | 34.35 | 0.975 |
| Hotdog 1/4x | 29.79 | 0.929 | 32.44 | 0.953 | 32.49 | 0.937 | 32.55 | 0.956 |
| Hotdog 1/8x | 28.72 | 0.939 | 32.28 | 0.940 | 31.87 | 0.929 | 32.15 | 0.956 |
| Materials 1x | 23.67 | 0.901 | 22.17 | 0.902 | 25.13 | 0.924 | 24.92 | 0.956 |
| Materials 1/2x | 25.70 | 0.946 | 24.17 | 0.915 | 28.23 | 0.935 | 27.85 | 0.972 |
| Materials 1/4x | 26.70 | 0.928 | 25.46 | 0.924 | 27.01 | 0.913 | 26.72 | 0.951 |
| Materials 1/8x | 26.01 | 0.806 | 24.47 | 0.842 | 27.68 | 0.860 | 26.62 | 0.863 |
| Mic 1x | 30.53 | 0.981 | 29.67 | 0.979 | 30.55 | 0.983 | 30.12 | 0.979 |
| Mic 1/2x | 34.02 | 0.972 | 31.72 | 0.973 | 33.98 | 0.975 | 33.52 | 0.976 |
| Mic 1/4x | 34.69 | 0.965 | 33.37 | 0.954 | 35.95 | 0.957 | 34.60 | 0.957 |
| Mic 1/8x | 35.54 | 0.958 | 36.00 | 0.954 | 35.35 | 0.953 | 35.52 | 0.952 |
| Ship 1x | 28.30 | 0.878 | 24.99 | 0.789 | 29.74 | 0.795 | 24.35 | 0.795 |
| Ship 1/2x | 29.74 | 0.901 | 27.02 | 0.812 | 27.11 | 0.899 | 27.17 | 0.829 |
| Ship 1/4x | 30.40 | 0.910 | 28.54 | 0.854 | 30.70 | 0.914 | 27.41 | 0.828 |
| Ship 1/8x | 31.42 | 0.931 | 29.00 | 0.892 | 31.40 | 0.891 | 28.17 | 0.888 |

We follow a similar training procedure to BACON [13]. Training data consists of sampled locations from the zero level set, where Laplacian noise is added for each point, as in [6]. As noted in BACON, the width of the Laplacian has a large performance impact, and so we use the same coarse and fine sampling procedure of BACON, where "fine" samples are produced using a small variance of $\sigma_L^2 = $ 2e-6 and "coarse" samples with $\sigma_L^2 = $ 2e-2. Samples are drawn in the domain $[-0.5, 0.5]^3$ and the following loss is used: $\lambda_{\text{SDF}}\|\mathbf{y}^c - \mathbf{y}_{\text{GT}}^c\|_2^2 + \|\mathbf{y}^f - \mathbf{y}_{\text{GT}}^f\|_2^2$, where $\mathbf{y}$ is the generated output, $\mathbf{y}_{\text{GT}}$ is the ground truth value, the $f$ and $c$ indicate fine and coarse samples. As in BACON, $\lambda_{\text{SDF}}$ is set to 0.01 for all experiments. SIREN and BACON are used as baselines, each trained for $200,000$ with a batch size of $5000$ coarse and $5000$ fine samples. The same optimization as for neural radiance field is used here.

Table 2: 3D shape fitting. CD is Chamfer Distance ($\times 10^6$). FS stands for F-score and NC stands for normal consistency.

| | #Iters | Armadillo CD | NC | FS | Dragon CD | NC | FS |
|---|---|---|---|---|---|---|---|
| SIREN | | 2.86 | 99.16 | 99.92% | 11.9 | 97.03 | 99.43% |
| BACON | 200k | 2.86 | 99.12 | 99.91% | 9.57 | 96.94 | 99.65% |
| BACON-last | | 2.88 | 99.05 | 99.91% | 5.11 | 96.92 | 99.67% |
| BACON-mean | | 2.86 | 99.07 | 99.91% | 9.80 | 96.83 | 99.57% |
| | 200k | 2.87 | 99.14 | 99.91% | 1.72 | 97.02 | 99.99% |
| | 100k | 2.86 | 99.14 | 99.92% | 1.73 | 96.99 | 99.99% |
| Ours | 60k | 2.87 | 99.13 | 99.91% | 1.74 | 96.93 | 99.99% |
| | 40k | 2.88 | 99.11 | 99.90% | 1.76 | 96.84 | 99.99% |
| | 20k | 2.91 | 99.05 | 99.91% | 1.82 | 96.67 | 99.99% |
| Oracle | | 2.85 | 99.15 | 99.92% | 1.67 | 97.84 | 100.00% |

| | #Iters | Lucy CD | NC | FS | Thai CD | NC | FS |
|---|---|---|---|---|---|---|---|
| SIREN | | 18.7 | 98.00 | 99.74% | 2.61 | 94.71 | 99.96% |
| BACON | 200k | 18.1 | 97.79 | 99.82% | 2.60 | 94.57 | 99.97% |
| BACON-last | | 4.95 | 97.03 | 99.73% | 2.60 | 94.59 | 99.97% |
| BACON-mean | | 22.1 | 97.94 | 99.94% | 2.61 | 94.42 | 99.97% |
| | 200k | 1.75 | 97.89 | 100.00% | 2.61 | 94.56 | 99.97% |
| | 100k | 1.75 | 97.87 | 100.00% | 2.60 | 94.57 | 99.96% |
| Ours | 60k | 1.76 | 97.81 | 100.00% | 2.64 | 94.45 | 99.96% |
| | 40k | 1.77 | 97.78 | 100.00% | 2.67 | 94.30 | 99.97% |
| | 20k | 1.81 | 97.66 | 100.00% | 2.75 | 93.95 | 99.96% |
| Oracle | | 1.71 | 98.54 | 99.99% | 2.54 | 95.55 | 99.97% |

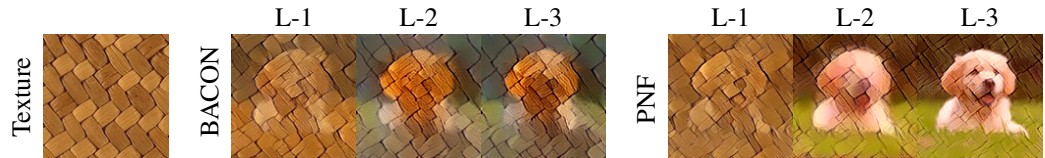

Figure 5: Additional texture transfer results. We optimize specific layers of the neural field. L-1 (layers 1-4), L-2 (layers 2-4), L-3 (layers 3-4).

## 3.2 Texture Transfer

For texture transfer, a random cat image is used (free-use licence) and texture image is used from [7]. As noted in the main text, we query the network on a $128^2$ grid and apply either (a) Content and style loss objectives as given in [10], or (b) Text-based texture manipulation objectives as given by CLIPStyler [12]. For baseline comparison, the same objectives are applied directly on the cat image, and the cat image pixel values are directly optimized.

For (a), we use the same objectives (content and style loss), loss weights and optimizer as used in the public code of https://pytorch.org/tutorials/advanced/neural_style_tutorial.html and [10]. Similarly, for (b), we similarly use the same we objectives, loss weights and optimizer as used in the official code of CLIPStyler [12] in https://github.com/cyclomon/CLIPstyler. Additional texture transfer corresponding on the Fig. 5(a) is given in Fig. 5.

## 3.3 Scale-space Representation

In this section, we will first show the derivation of the PNF used for the scale-space representation. Then we will provide detailed description of the experiment and provide numerical results.

### 3.3.1 Derivation of Scale-space Fourier PNF

Suppose the signal of interest can be represented by Fouier bases as $g(\mathbf{x}) = \sum_n \alpha_n \exp\left(\omega_i^T \mathbf{x}\right)$, then we know analytically the Gaussian convolved version should be $f(\mathbf{x}, \Sigma) = \sum_n \alpha_n \exp(\omega_i^T \Sigma \omega_i) \exp(i\omega_i^T \mathbf{x})$. If we assume that our Fourier PNF can learn the ground truth representation well, then one potential way to achieve this is setting $\gamma(\mathbf{x}, \Sigma)_n = \exp(-\frac{1}{2}\omega_n^T \Sigma \omega_n) \exp(-i\omega_n^T \mathbf{x})$. Doing this naively with Fourier PNF only yields a bad approximation, since each coefficient is off by an error term $E(n, I_n)$ from the ground truth. We will derive such error term in detail below.

For a particular $F_k$ (Equation (4) in the main text), let $\tilde{F}_k(\mathbf{x}, \Sigma)$ be the output of naively replacing the Fourier basis encoding $\gamma$ with the intergrated Fourier basis encoding: $\gamma(\mathbf{x}, \Sigma)$. Then we have:

$$\mathbb{E}_{\mathbf{x} \sim \mathcal{N}(\mu, \Sigma)} [F_k(\mathbf{x})] \tag{68}$$

$$= \sum_n \alpha_n \exp\left(-\frac{1}{2}\omega_n^T \Sigma \omega_n^T\right) \exp(i\omega_n^T x) \tag{69}$$

$$= \sum_n \exp\left(-\frac{1}{2}\omega_n^T \Sigma \omega_n^T\right) \sum_{I \in I_n} \alpha_{nI} \exp\left(i \left(\sum_{j \in I} \omega_j\right)^T \mathbf{x}\right) \tag{70}$$

$$= \sum_n \sum_{I \in I_n} \exp\left(-\frac{1}{2}\omega_n^T \Sigma \omega_n^T\right) \alpha_{nI} \exp\left(i \left(\sum_{j \in I} \omega_j\right)^T \mathbf{x}\right) \tag{71}$$

$$= \sum_{n, I \in I_n} \alpha_{nI} \exp\left(-\frac{1}{2} \sum_{j \in I} \omega_j^T \Sigma \omega_j^T\right) \exp\left(-\frac{1}{2} \sum_{j \neq l, j, l \in I} \omega_j^T \Sigma \omega_l^T\right) \exp\left(i \left(\sum_{j \in I} \omega_j\right)^T \mathbf{x}\right) \tag{72}$$

$$= \underbrace{\sum_{n, I \in I_n} \alpha_{nI} \left(\prod_{l \in I} \exp\left(-\frac{1}{2}\omega_l^T \Sigma \omega_l^T + i\omega_n^T \mathbf{x}\right)\right)}_{\text{PNF output:} \tilde{F}_k(\mathbf{x}, \Sigma)} \underbrace{\exp\left(-\frac{1}{2} \sum_{l, j \in I, l \neq j} \omega_l^T \Sigma \omega_j^T\right)}_{\text{Error terms } E_k(n, I_n)}, \tag{73}$$

where $I_n$ include all indexes to choose one basis per $\gamma_j$ layer (as defined in Equation (6) of the main text) such that $\forall I \in I_n, \sum_{l \in I} \omega_l = \omega_n$. We use $\alpha_{nI}$ to denotes the coefficients gathered along the index of $I$. We can compute these coefficients analytically by applying similar analysis as MFN [9]. The formua above shows that $\tilde{F}_k(\mathbf{x}, \Sigma)$ is not a good approximation since each term $\alpha_{n,I}$ is off by different factor $E_k(n, I_n)$.

Table 3: Numerical results for scale-space interpretation experiments. IPE: intergrated positional encoding [2]; RIPE: intergrated positional encoding with randomly fourier features; IPE-sup: supervised IPE with both 1x and 1/4x resolution. While PNF is only supervised with 1x, it's capable of inerpolating into smaller scale without breaking the image structure (as shown in good SSIM). This is more impressive as the basis functions for PNF is sampled from random directions (i.e. not necessarily aligned with the eigenvectors directions of the test-time Gaussian covariances). Doing the same thing to IPE results in worse performance as shown in RIPE.

| Model | PSNR | | | | SSIM | | | |
|---|---|---|---|---|---|---|---|---|
| | 1x | 1/2x | 1/4x | 1/8x | 1x | 1/2x | 1/4x | 1/8x |
| RIPE | 39.41 | 6.35 | 6.54 | 6.63 | 96.95 | 20.43 | 22.73 | 23.8 |
| IPE | 37.58 | 24.62 | 21.94 | 14.68 | 94.67 | 73.88 | 58.51 | 37.95 |
| BACON | 40.63 | 8.88 | 7.19 | 7.2 | 97.39 | 43.11 | 28.16 | 29.66 |
| PNF | 36.45 | **27.07** | 29.74 | **24.6** | 95.98 | **86.30** | 83.33 | **72.66** |
| RIPE-sup | **50.11** | 11.91 | 26.64 | 10.69 | **99.62** | 15.51 | 73.24 | 11.27 |
| IPE-sup | 34.76 | 26.8 | **30.00** | 18.41 | 90.97 | 84.81 | **89.65** | 65.7 |

Fortunately, we know that all bases chosen from $I_n$ must come from a specific set of that shares the same angle (i.e. the limiting-subbands all shared a direction vector $\mathbf{d}$ and an angular width $\gamma$ by the definition of $F_i$), but with different range of for the norm $\|\omega\|_p$. This suggests that we can approximate the error terms in the following way:

$$E_i(\cdot, I) \approx \exp\left(-\frac{1}{2}\mathbf{d}_I^T \Sigma \mathbf{d}_I^T \sum_{k,l \in I, k \neq l} \bar{r}_k \bar{r}_l\right) = A_i(\Sigma), \tag{74}$$

where $\bar{r}_k, \bar{r}_l$ are the medium radius of each subband within the subband series $\mathcal{S}_i$. Specifically, $\bar{r}_l = \frac{1}{2}(\alpha_l + \beta_l)$ if the $l^{th}$ subband used for $F_k$ is $S_l \in \mathcal{S}_i$ lower- and upper- bounded by $\alpha_l$ and $\beta_l$. This way, the error estimation only depends on the PNF-controllable set of subbands used to developed the network $F_k$, but not the network parameter. This means we can modify the network architecture of each $F_i$ in test-time to compute $A_k(\Sigma)$ to approximate error term $E_k(\cdot, I)$:

$$\mathbb{E}_{\mathbf{x} \sim \mathcal{N}(\mu, \Sigma)}[F_k(\mathbf{x})] \tag{75}$$

$$\approx \sum_{n, I \in I_n} \alpha_{nI} \left(\prod_{l \in I} \exp\left(-\frac{1}{2}\omega_l^T \Sigma \omega_l^T + i\omega_n^T \mathbf{x}\right)\right) A_k(\Sigma) \tag{76}$$

$$= A_i(\Sigma) \sum_{n, I \in I_n} \alpha_{nI} \sum_{n, I \in I_n} \left(\prod_{l \in I} \exp\left(-\frac{1}{2}\omega_l^T \Sigma \omega_l^T + i\omega_n^T \mathbf{x}\right)\right) \tag{77}$$

$$= A_i(\Sigma) \tilde{F}_k(\mathbf{x}, \Sigma) \tag{78}$$

With this, our scale-space interpolation Fourier PNF can be written as sums of outputs of all subband series $\mathcal{S}_k$:

$$\tilde{F}(\mathbf{x}, \Sigma) = \sum_k A_k(\Sigma) \tilde{F}_k(\mathbf{x}, \Sigma). \tag{79}$$

This approximation will be exact if $\gamma = 1$ and $\alpha = \beta$ for all subbands. But it also means we need infinite number of $F_k$ to tile the space. The smaller $\gamma$ is, the more error can this potentially occurs. Similarly, the larger $|\alpha - \beta|$ is, the more error this approximate can creates. This shows a trade-off between compute and the approximation error.

### 3.3.2 Details

The image used for Scale-space Representation is taken from the Set12 dataset [17] and downsampled to $128^2$ resolution. We train all models for 1500 iterations with early stopping when the training PSNR reaches 40. All models use Adam optimizer with learning rate $1e - 3$. The ground truth is

generated from applying Gaussian filter of size $3 \times 3$ (for 1/2x), $5 \times 5$ (for 1/4x), and $65 \times 65$ (for 1/8x). We compute the ground truth using OpenCV [3]. For $\Sigma$, we use anisotropic covariance matrix of the form $\Sigma = \sigma^2 I$ with $I$ being the $2 \times 2$ identity matrix. We set $\sigma^2 = 1$ for 1x, $\sigma = 2$ for 1/2x, $\sigma = 4$ for 1/4x, and $\sigma = 8$ for 1/8x. For PNF, we used the spherical tiling $\mathcal{T}_{rect}$ with 10 subbands with 2-times overcomplete. So each subband has angular width of $\frac{2}{12}$ as we need to restrict the subband into either total-vertical or total-horizontal area (i.e. $R_\infty(1, +1)$ or $R_\infty(0, +1)$ as definted in Definition 1.7). The band limits for each intermediate residual output are set to be the following fraction of the Nyquist rates: $[0, \frac{1}{8}]$, $[\frac{1}{16}, \frac{1}{4}]$, $[\frac{1}{8}, \frac{1}{2}]$, and $[\frac{1}{4}, 1]$. In this experiment, we compare to the following baselines:

1. RIPE: we use intergrated random Fourier features in the following form: $\exp\left(-\frac{1}{2}\omega^T \Sigma \omega\right) \sin(\omega^T \mathbf{x})$. These intergrated random Fourier features will be followed by a ReLUMLP.

2. IPE: we use ReLUMLP on axis-aligned intergrated positional encodings as shown in [2].

3. BACON: replacing BACON's filter with the intergrated random Fourier Features.

4. RIPE-sup: same as RIPE, but supervised with both 1x and 1/4x.

5. IPE-sup: same as IPE, but supervised with both 1x and 1/4x.

All baselines have about 0.3M parameters (which is about the same number of floating points as the 128x128 image). All models are trained on 1x images except for IPE-sup, which is also supervised for 1/4x. The results are shown in Tab. 3. As we can see from the table, PNF is capable of representing 1/2x, 1/4x, and 1/8x resonably well while supervised only on 1x resolution. Comparing to IPE-sup, PNF is able to match the performance at 1/4, eventhough PNF is not supervised at 1/4x. Moreover, PNF used random Fourier features instead of the axis-align basis functions. As shown in the comparison between RIPEs and IPEs, axis-align positional encoding performs better in this setting, which we hypothesize the reason being that during the test-time the $\Sigma$'s eigenvectors are also axis-align. PNF model choose direction randomly within the angle of the PNF-Controllable set of subbands, and it still achieves very strong performance.

## 3.4 Licenses

DIV2K [1] dataset is available for academic use only. The multiscale Blender dataset [2] is provided under the creative commons public license. The Stanford 3D scanning repository[4] dataset is available to use for research purposes. Data used for texture transfer is available for academic use or is free to use. Set12 dataset [17] is available under under the creative commons public license. All data does not contain personally identifiable information or offensive content.

## 3.5 GPU Resources

Image based experiments require a single GPU such as an NVIDIA Titanx/Titanrtx GPU. For Neural fields and 3D Signed Distance Fields, a higher GPU number is required per experiment, and up to 4 NVIDIA Titanrtx GPUs. An internal GPU cluster was used for experiments.

## 3.6 Negative Societal Impact

On some experimental settings, our framework requires a large GPU and memory requirement, which may have an adverse environmental impact. Neural fields have been used in a verity of generative settings, our framework may be used in potential negative use of such generative models. For instance, for neural radiance fields, creating realistic novel views of fake scenes, could be used maliciously. Such cases could be better handled by developing tools to detect fake scenes, or any other content generated using neural fields such as our PNF.

---

[4] http://graphics.stanford.edu/data/3Dscanrep/