# OpenReview forum: "Polynomial Neural Fields for Subband Decomposition and Manipulation"
_NeurIPS.cc/2022/Conference — NeurIPS 2022 Accept_

### Official Review · Reviewer_pc1o · 2022-07-11

**Rating:** 7
**Confidence:** 3
**Soundness:** 3 good
**Presentation:** 3 good
**Contribution:** 3 good

**Summary:**

The paper proposes an approach to allow subband manipulation in neural fields. Specifically, subbands are decomposed in the Fourier space and there is one network for each subband and the final results are combined. Subband manipulation is achieved via manipulating the corresponding neural network.

**Questions:**

Line 99: how does $\gamma_i$ relate to $f$? The statement is not clear.
Line 110: What are $a_i$, $b_i$ and $I$? The statement is not clear.

**Limitations:**

Memory limitation is mentioned and the societal impact is adequately addressed.

**Strengths And Weaknesses:**

## Strengths
1. The idea is interesting and it adds a subband manipulation capability to neural fields. I believe this is a valuable contribution.
2. The proposed approach yields marginally better results than existing approaches.
3. The subband manipulation capability is demonstrated via applications such as texture transfer.

## Weaknesses
1. In this approach, the subband decomposition has to be predetermined and having one network per subband increases the memory complexity of the overall approach. Due to this, the manipulation capability might be limited. Furthermore, how does the number of parameters of the proposed method compares to an existing approach like BACON?

## Post rebuttal
Thank you for providing the parameter efficiency analysis. I'm already positive and increasing my score.

---

> ### Author Response · Authors · 2022-08-02
> **Thank you for your review!**
>
> We thank the reviewer for the insightful review! We are glad that the review found our paper making a valuable contribution by adding subband manipulation capability to neural fields.
>
>
> **Predetermined subband decomposition.**
> We agree with the reviewer that our method needs a subband decomposition defined before training. The use of a predetermined subband decomposition dates back to classical wavelet decomposition as well as Laplacian [8] and Steerable [51] pyramids. These subband decompositions have been instrumental in many applications such as texture analysis and synthesis. One of our main contributions is to enable such decomposition for neural fields and we demonstrate such a technique is useful in a variety of applications (Section 4).
>
> Alleviating the need of predeterministic subband decomposition is an interesting future direction. One potential way to achieve this is to design a basis function to enable the network to learn tunable subband decompositions. We believe that our work lays the basic theories for achieving this goal and hope to see future research in this direction.
>
> **Memory complexity.**
> It’s true that we use one network to encode one subband series and the final output is an ensemble of the outputs from these networks. We mentioned in the limitation section that the activation memory will thus increase linearly to the number of subband series.
>
> PNF can still be very compact in terms of the storage memory. This is due to the fact that each small network only needs to capture signals from a subband instead of the whole signal, which is arguably an easier task that requires much less network parameters to learn. Please refer to our general comments for a detailed discussion about parameter efficiency.
>
> **Clarifications.** Thanks for pointing out the potential confusions in the paper!
>
> Line 99: $\gamma$ is a function that takes $\mathbb{R}^n$ coordinate and maps it to a $d$-dimensional vector. We denote each dimension of the output vector $\gamma(x)$ is denoted as $\gamma_i$.
>
> Line 110:  $b_1, b_2$ are two basis functions from family $\mathcal{B}$. $a_i$ are complex coefficients. The property requires that for each pair of basis functions, there exists a series of coefficients that can express their product as a weighted sum of the basis.
>
> We will make the math notations more clear in the revision.

---

### Official Review · Reviewer_kNCf · 2022-07-11

**Rating:** 8
**Confidence:** 3
**Soundness:** 4 excellent
**Presentation:** 4 excellent
**Contribution:** 3 good

**Summary:**

The paper describes a novel class of neural fields, which have explicit control over their frequency content. Inspired by classical wavelet decomposition of signals, the approach allows to construct neural fields that covers the Fourier spectrum in separate non-overlapping sections. Each section is controlled by both a lower and upper band-limited, as well as an angular sensitivity. Until now, NFs where only controllable via an upper frequency band-limit. The presented NFs thus have nice properties which could benefit a wide range of applications for which frequency content control is important (of which several are addressed in this paper), the NFs also seem to have generally good approximation quality (both in terms of quality of fit as well as convergence speed).

**Questions:**

None

**Limitations:**

I see no issues here.

**Strengths And Weaknesses:**

**Strengths**
* The paper is well-written
* The paper is timely
* The paper solves a relevant and so far unsolved problem for controlling frequency content in NFs
* The paper is sound and is thorough in its citations (even to classic works like Simoncelli-Freeman)

**Weaknesses**
* A discussion on parameter efficiency and computational efficiency would be appreciated

**Literature comment**
In terms of literature that band-limits NFs from above it only makes mention of BACON, however [R21], which is on controlling frequency content (band-limiting) via MFNs of the Gabor type, predates BACON (probably independently solved the same problem?)
[R21] Romero, David W., et al. "FlexConv: Continuous Kernel Convolutions With Differentiable Kernel Sizes." ICLR. 2021.

---

> ### Author Response · Authors · 2022-08-02
> **Thanks for your review!**
>
> We thank the reviewer for the positive and supportive feedback and for appreciating the strengths of our work.
>
> **Parameter and Computational efficiency**
> Please refer to the general comments for a detailed discussion of computational and parameter efficiency. In short, PNF can be as compact as the baselines (i.e. using about the same amount of parameters) while achieving similar performance. The training and inference time is longer per step due to large activation memory (as mentioned in the limitation section). We believe that this can be addressed by optimizing the implementation. Despite slower training/inference time, PNF still converges faster in wall time.
>
> **Literature comment**
> We thank the reviewer for noting the work of [R21], which we will gladly add in the paper. As noted by the reviewer, [R21] considers the ability to control frequency from above via MFNs of the Gabor type. Our framework handles a more general set of basis functions and network topologies and can band-limit a signal also from below, thus enabling decomposition of a signal’s frequency bands.

---

### Official Review · Reviewer_9C3L · 2022-07-15

**Rating:** 5
**Confidence:** 3
**Soundness:** 3 good
**Presentation:** 2 fair
**Contribution:** 3 good

**Summary:**

The authors propose a basis-encoded polynomial neural fields with a basic theoretical framework for implicit signal manipulation. Specifically, a set of PNFs (e.g., a Fourier PNF), with the outputs be decomposed in a fine-grained manner in the frequency domain. Experiments show that PNFs achieve comparable performance with other SOTA methods.

**Questions:**

1. Lack of efficiency analysis, how about the training and inference time with compared methods.
2. Why only perform experiments on Blender scenes for Neural Radiance Field?
3. More related works for Polynomial Neural Networks.

**Ethics Review Area:**

["I don’t know"]

**Limitations:**

Please refer to the Questions section.

**Strengths And Weaknesses:**

1. An interpretable way for signal manipulation.
2. Multiple manipulation applications to verify the proposed ideas.
3. With several theories to support the proposed ideas.

---

> ### Author Response · Authors · 2022-08-02
> **Thanks for the review!**
>
> We thank the reviewer for the overall positive and constructive feedback.
>
> **Efficiency analysis**
> Please refer to our general comment for a detailed discussion of parameter and computation efficiency analysis. In brief, PNF achieves comparable performance with SOTA methods using a similar number of parameters. PNF takes longer training/inference time per step since it requires larger activation memory. Nonetheless, PNF still can achieve faster convergence time since it takes much fewer steps to converge.
>
> **NeRF experiments on Real-world Scene.**
> Our method is applicable to fitting NeRF for real-world scenes. We follow the prior work of BACON for the experiment set-ups and BACON focuses on Blender scenes. Note the main contribution of the paper is a novel neural fields architecture that allows provable decomposition and manipulation in the granularity of subbands. The NeRF experiment shows that our method is capable of achieving such decomposition and manipulation without sacrificing expressivity. Nevertheless, we will attempt to add real-world scene comparisons in the next revision.
>
> **Related works for Polynomial Neural Networks (PNNs).**
> We discussed polynomial neural networks in L81-L84 and L48-50. Basis-encoded Polynomial neural fields can be thought as evaluating a polynomial neural network with a selected set of basis functions. This formulation is more general than the original PNNs, which can be thought of as using a polynomial function basis : $\{b_i(x) = x^i\}_{i=1}^{\infty}$. Our work studies a variety of different basis functions including the Fourier basis (in the main paper) and the Gabor basis (in the supplementary). The understanding from PNNs can potentially be extended to PNFs and we believe that our analysis can also be helpful for the PNN community.
>
> In the revision, we will include additional citations (for example, [A] and [B]) of recent PNNs and provide detailed discussion of polynomial neural networks in our related work section.
>
> [A] Kileel, Joe, Matthew Trager and Joan Bruna. “On the Expressive Power of Deep Polynomial Neural Networks.” NeurIPS (2019).
> [B]Choraria, M., Dadi, L., Chrysos, G.G., Mairal, J., & Cevher, V. (2022). The Spectral Bias of Polynomial Neural Networks. ICLR 2022.

---

### Official Review · Reviewer_k1EA · 2022-07-16

**Rating:** 7
**Confidence:** 3
**Soundness:** 3 good
**Presentation:** 3 good
**Contribution:** 3 good

**Summary:**

This work proposes to build neural field network using polynomial basis and a Fourier extension. Their goal is to provide sub-band frequency control for the output of the network, but for both upper and lower band frequencies, instead of only upper band frequencies per layer [1] . The authors propose to concatenate finite degree multivariate polynomials which facilitate sub-band decomposition. The final network operates with a controllable set of sub-band decompositions as it constitutes an ensemble across networks with different sub bands. The authors show the performance of the network on multiple types of tasks coordinate networks are used, including pixel reconstruction with different levels of noise, 3d shape reconstruction. The performance of the network is qualitatively better than previous work in these reconstructions for high frequency components, including [1], but quantitatively comparable.

[1] Lindel et al. 2022. BACON

**Questions:**

- It is unclear to me from the text, which parameters beyond the network weights need to be optimized?
- What is the cost of scaling this approach with the dimensions of the image?
- The results in Figs 2, 4 and, 5 show results that appear to be qualitatively superior than previous work including [1]; however, the quantitative results in Tables 1,2,3 do not provide any confidence intervals, and are .## significant digits. What are the limitations of the metrics used in Tables 1,2 and 3?

**Strengths And Weaknesses:**

This work leverages signal processing techniques for subband decomposition to build better coordinate networks. This in turn leads to an interpretable coordinate network where the layers are lower and upper banded. One interesting result from this work is Fig
4, where the results show the network can isolate the image content and apply different filters at different layers.

The paper reads as a story on how to build a network Fig 1b, this in turn makes it hard to read unless you are familiar with all the component necessary for subband decomposition and coordinate networks. The paper would be accessible to a wider audience if it builds from the components/architecture necessary/standard for coordinate networks and the new components proposed by this work.
The results in Figs 2, 4 and, 5 show results that appear to be qualitatively superior than previous work including [1]; however, the quantitative results in Tables 1,2,3 do not provide any confidence intervals, and are .## significant digits.
The

[1] Lindel et al. 2022. BACON

---

> ### Author Response · Authors · 2022-08-02
> **Thanks for your positive feedback!**
>
> We thank the reviewer for the positive feedback and for appreciating the strengths of our work.
>
> **Writing.**
> We appreciate the clarity suggestion. We will add a subsection in Section 3 and introduce some of the related widgets of coordinate-based neural network (e.g. Random Fourier embedding, MFN, BACON) and how to use them to build networks like Figure 1(b).
>
> **Optimized parameters**
> Only the network weights are optimized, corresponding to the $W$’s in Eq.4-Eq.6. Other parameters such as the initialization schema of the basis encoding function $\gamma$ are considered hyper-parameters.
>
> **Scaling cost with respect to image resolution.**
> To train Fourier PNF on images with 2x resolution, one can potentially use the same network architecture (i.e. same amount of parameters), but double the initialization frequency when constructing $\gamma$. But we usually found that increasing the width of the network is beneficial when fitting more complicated signals. For example, increasing the width of the network by $10%$ is sufficient to gain comparable results for the image overfitting experiment of camera men when scaling from $256^2$ to $512^2$. Note that such an increase in the number of parameters is also required in prior work like BACON.
>
> The forward and backward computation (i.e. time and memory) is linear to the number of input points, so we expect a 4x amount of the original compute.
>
> ***Limitation of metrics, confident interval, and significant digits.**
> We follow the prior work (BACON) to use PSNR, SSIM, and CD to evaluate the neural field’s ability to overfit a signal. Sometimes these metrics are not always well-correlated with human perception [A, B]. We will include confidence intervals and significant digits in the paper revision.
>
> [A] Zhang, R., Isola, P., Efros, A.A., Shechtman, E., & Wang, O. (2018). The Unreasonable Effectiveness of Deep Features as a Perceptual Metric. 2018 IEEE/CVF Conference on Computer Vision and Pattern Recognition, 586-595.
> [B]Smirnov, D., Fisher, M., Kim, V.G., Zhang, R., & Solomon, J.M. (2020). Deep Parametric Shape Predictions Using Distance Fields. 2020 IEEE/CVF Conference on Computer Vision and Pattern Recognition (CVPR), 558-567.

---

### Author Response · Authors · 2022-08-02
**General replies to reviewers**

We would like to thank the reviewers for the unanimously positive reviews! We are glad that the reviewers found our paper “timely”, “sound”, and “well-written”. In this paper, we propose a novel neural field architecture called Basis-encoded Polynomial Neural Fields (PNF). To the best of our knowledge, our method is the first neural field that’s capable of achieving analytical subband decomposition of signals. In addition, our paper also proposes a set of theories to generalize the PNF architecture to different basis functions and network topologies. We demonstrate the benefits of PNF on a variety of tasks including texture synthesis and scale-space interpolation. We hope that our paper can inspire new designs of neural fields that are both expressive and interpretable. Next we address the reviewers’ questions regarding efficiency analysis.

--------

## Parameter efficiency
Reviewer 9C3L, kNCf, and pc1o inquire about the parameter efficiency of PNF. We choose the hyperparameters for our method (e.g. hidden layer size) so as to have a comparable number of parameters for all default configurations of the baselines. With a comparable number of network parameters, we are able to build neural fields with expressivity on par with the SOTAs while allowing analytical subband manipulation. The following table shows the number of parameters in our model in comparison with that of the baselines in the expressivity experiments. We will include this parameter analysis in the revision.

| Expriment | PNF | BACON | RFF | SIREN |
|-|-|-|-|-|
| NeRF 1x | 0.46M | 0.54M  | N/A | N/A |
| NeRF 1/2x | 0.34M | 0.41M  | N/A | N/A |
| NeRF 1/4x | 0.23M | 0.27M  | N/A | N/A |
| NeRF 1/8x  | 0.12M | 0.14M  | N/A | N/A |
| Image | 0.28M | 0.27M  | 0.26M | 0.26M  |
| SDF | 0.59M | 0.54M  | N/A | 0.53M  |

--------

## Training and inference time.
As per the reviewer 9C3L and knCf’s requests, we profile the training and inference time of the image experiment in the following tables for the cameramen image. As mentioned in our limitation section (L305-307), PNF requires forwarding through an ensemble of subband networks, and thus requires more activation memory. This usually results in longer training/inference time per step when implemented without any optimization. We believe that this limitation can potentially be addressed by exploiting hardware’s parallelism ability (e.g. writing customized CUDA kernel). Nonetheless, our model converges in very few steps, which leads to faster overall convergence time as shown in the following table.

| - | Time(s)/Step | Time(s) to 36 PSNR | Final PSNR/SSIM (5k steps) |
|-|-|-|-|
| BACON | 0.16 | 177 | 37.45/97.33 |
| PNF | 0.64 | 96 | 37.45/97.44 |
| SIREN | 0.1| 163 | 36.9/97.50 |
| RFF | 0.08 | 275 | 36.23/95.05 |

---

### Meta-Review · Area_Chair_36iu · 2022-08-21

**Recommendation:** Accept
**Confidence:** Certain

**Metareview:**

There is a clear consensus for accepting the paper. The area chair agrees with the reviewer's comments and follows their recommendation.

**Award:**

No

---

### Decision · Program_Chairs · 2022-09-14

Accept